# Implementation of HONO into the chemistry-climate model CHASER (V4.0): roles in tropospheric chemistry

Phuc T. M. Ha[1], Yugo Kanaya[2], Fumikazu Taketani[2], Maria Dolores Andrés Hernández[3], Benjamin Schreiner[4], Klaus Pfeilsticker[4], Kengo Sudo[1,2]

[1] Graduate School of Environmental Studies, Nagoya University, Nagoya 464-8601, Japan
[2] Research Institute for Global Change, JAMSTEC, Yokohama 236-0001, Japan
[3] Institut für Umweltphysik, Universität Bremen, Otto-Hahn-Allee 1, Bremen 28359, Germany
[4] Institut für Umweltphysik (IUP), Universität Heidelberg, INF 229, Heidelberg 69120, Germany

*Correspondence to*: Phuc T. M. Ha (hathiminh.phuc@gmail.com)

**Abstract.** Nitrous acid (HONO) is an important atmospheric gas given its contribution to the cycles of $NO_x$ and $HO_x$, but its role in global atmospheric photochemistry is not fully understood. This study implemented three pathways of HONO formation in the chemistry-climate model CHASER (MIROC-ESM) to explore three physical phenomena: gas-phase kinetic reactions (GRs), direct emission (EM), and heterogeneous reactions on cloud/aerosol particles (HRs). We evaluated the simulations by the atmospheric aircraft-based measurements from EMeRGe-Asia-2018 (Effect of Megacities on the Transport and Transformation of Pollutants on the Regional to Global scales), ATom1 (atmospheric tomography), observations from the ship *R/V Mirai*, EANET (Acid Deposition Monitoring Network in eastern Asia) / EMEP (European Monitoring and Evaluation Programme) ground-based stationary observations, and the OMI (Ozone Monitoring Instrument). We showed that the inclusion of the HONO chemistry in the modeling process reduced the model bias against the measurements for $PM_{2.5}$, $NO_3^-/HNO_3$, $NO_2$, OH, $HO_2$, $O_3$, and CO, especially in the lower troposphere and the North Pacific (NP) region.

We found that the retrieved global abundance of tropospheric HONO was 1.4 TgN. Of the three source pathways, HRs and EM contributed 63% and 26% to the net HONO production, respectively. We also observed that, reactions on the aerosol surfaces contributed larger amounts of HONO (51%) than those on the cloud surfaces (12%). The model exhibited significant negative biases for daytime HONO in the Asian off-coast region, compared with the airborne measurements by EMeRGe-Asia-2018, indicating the existence of unknown daytime HONO sources. Strengthening of aerosol uptake of $NO_2$ near surface and in the middle troposphere, cloud uptake, and direct HONO emission were all potential yet-unknown HONO sources. The most promising daytime source for HONO found in this study was the combination of enhanced aerosol uptake of $NO_2$ and surface-catalyzed $HNO_3$ photolysis (maxST+JANO3-B case), which could also remedy the model bias for $NO_2$ and $O_3$ during EMeRGe. We also found that the simulated HONO abundance and its impact on $NO_x$-$O_3$ chemistry were sensitive to the yield of the heterogeneous conversion of $NO_2$ to HONO (vs. $HNO_3$).

Inclusion of HONO reduced global tropospheric $NO_x$ (NO + $NO_2$) levels by 20.4%, thereby weakening the tropospheric oxidizing capacity (OH, $O_3$) occurring for $NO_x$-deficit environments (remote regions and upper altitudes), which in turn,

increased CH₄ lifetime (13%) and tropospheric CO abundance (8%). The calculated reduction effect to global ozone level reduced the model overestimates for tropospheric column ozone against satellite OMI for a large portion of the North hemisphere. HRs on the surfaces of cloud particles, which have been neglected in previous modeling studies, were the main drivers of these impacts. This effect was particularly salient for the substantial reductions of levels of OH (40–67%) and O₃ (30–45%) in the NP region during summer, given the significant reduction of NOₓ level (50–95%). In contrast, HRs on aerosol surfaces in China (Beijing) enhanced OH and O₃ winter mean levels by 600–1700% and 10–33%, respectively, with regards to their minima in winter. Furthermore, sensitivity simulations revealed that the heterogeneous formation of HONO from NO₂ and heterogenous photolysis of HNO₃ coincided in the real atmosphere. Nevertheless, the global effects calculated in the combined case (enhancing aerosol uptakes of NO₂ and implementing heterogeneous photolysis of HNO₃), which most captured the measured daytime HONO level, still reduced the global tropospheric oxidizing capacity. Overall, our findings suggest that a global model that does not consider HONO heterogeneous mechanisms (especially photochemical heterogeneous formations) may erroneously predict the effect of HONO in remote areas and polluted regions.

## 1. Introduction

Nitrous acid (HONO) is an important atmospheric gas as it participates in the cycles of nitrogen oxides ($NO_x = NO + NO_2$) and radical chemistry (OH, HO₂, and RO₂) (Kanaya et al., 2007; Ren et al., 2013; Whalley et al., 2018). Researchers have suggested to include the HONO chemistry in atmospheric chemistry models for more accurate simulations of oxidative substances (Jacob, 2000; Li et al., 2011). Despite the empirical evidences have indicated that the HONO concentrations in urban environments can reach 14 ppbv at night and can reach several hundred pptv throughout the day (Appel et al., 1990; Febo et al., 1996; Kanaya et al., 2007; Lee et al., 2016; Tan et al., 2017; Whalley et al., 2018), the HONO formation mechanism remains unclear. More specifically, the mechanisms of the HONO daytime sources have recently attracted considerable attention of researchers (Kleffmann et al., 2003; Li et al., 2014; VandenBoer et al., 2013; Xue et al., 2021a,b; Ye et al., 2018).

The only homogeneous reaction known to produce HONO in the troposphere is the direct combination of OH and NO (R2). Note that the major loss of HONO occurs via photolysis (R1) in the atmosphere at 300–405 nm:

(R1) HONO + h$\nu$ → OH + NO (300 nm < λ < 405 nm)

(R2) NO + OH + M → HONO + M

(R3) HONO + OH → $NO_2$ + $H_2O$

Moreover, the photolysis of HONO (R1) has attracted considerable attention in the literature as a critical source of OH radicals in the polluted urban atmosphere (e.g., Calvert et al., 1994; Harris et al., 1982; Jenkin et al., 1988; Platt and Perner, 1980). The OH level at sunrise can be increased by a factor of 5 due to the photolysis of HONO, with the regional daily maximum O₃ level increasing by 8% (Jenkin et al., 1988). Besides the direct loss via photolysis, the reaction of HONO with OH (R3) may also contribute to the daytime loss of HONO (Burkholder et al., 1992).

Notably, some night-time measurements hinted on the heterogeneous sources of HONO from aerosol surfaces. For instance, Harrison and Kitto (1994) have provided evidence about the HONO source from high concentration episodes of > 10 ppbv $NO_2$ for grassland in eastern England (Harrison and Kitto, 1994). Two reactions have been widely suggested to produce HONO on aerosol surfaces: $2NO_2 + H_2O \rightarrow HONO + HNO_3$ and $NO + NO_2 + H_2O \rightarrow 2HONO$. The first process has been proven to be first-order with $NO_2$ and $H_2O$ in reaction chamber studies (Sakamaki et al., 1983, Jenkin et al., 1988). The second process was evaluated by using laboratory surfaces (Sakamaki et al., 1983, Jenkin et al., 1988) and by using field observations in the presence of high $O_3$ and when $NO_2$ was the dominant form of $NO_x$ (Kessler and Platt, 1984). As a result, the second process was proposed as a peculiarly important source of HONO in the urban atmosphere (Ammann et al., 1998; Gerecke et al., 1998). In the past two decades, researchers have investigated the heterogeneous $NO_2$ reactivity on vegetated, aqueous, sea salt, carbonaceous, and soot surfaces (Acker et al., 2001, 2006; Arens et al., 2001; Kleffmann and Wiesen, 2005; Kleffmann et al.,1998; Lammel and Cape, 1996; Lee et al., 2016; Notholt et al., 1992; Reisinger, 2000; Rubio et al., 2002; Stutz et al., 2002). In our model, these two processes are simplified as $NO_2 \rightarrow 0.5\ HONO + 0.5\ HNO_3$ (R4) and $NO_2 \rightarrow HONO$ (R5).

Also, some modeling studies have reported overestimations of HONO over remote areas, indicating the HONO release from or deposition in snow (Chu et al., 2000; Fenter and Rossi, 1996; Kerbrat et al., 2010), partitioning to cloud water (Bongartz et al. 1994; Cape et al., 1992; Harrison and Collins, 1998; Mertes and Wahner, 1995), and deliquescent aerosol surfaces (Harrison and Collins, 1998). The loss process occurs via the reaction $HONO + H_2O \rightarrow NO^- + H_3O^+$, simplified in our model as $HONO \rightarrow NO$ (R6) for surfaces of liquid and aqueous sulfate aerosols.

The natural sources of HONO include plant-foliar cuticles or soil biological crust (Hayashi and Noguchi, 2006; Oswald et al., 2013; Porada et al., 2019; Su et al., 2011), with estimated global total emission of 0.69 Tg $yr^{-1}$ HONO–N (Porada et al., 2019). Given the widespread occurrence of nitrite-fertilized soil in natural environment, highly acidic soils are arguably the strong sources of HONO and OH (Su et al., 2011). This potentially important source has been likely overseen by many previous modeling studies at both global and regional scales. Soil emissions could sustain the daytime HONO budget at relatively low aerosol concentrations (Lu et al., 2018). Anthropogenic activities can also directly emit HONO through incomplete combustion, as vehicles, for instance, can yield as high concentrations as 7 ppb (Kirchstetter et al., 1996; Kurtenbach et al., 2001). In regional air quality models, HONO sources from vehicles and vessels are often given at 0.8–2.3% of $NO_x$ emissions level, given the differences between gasoline and diesel vehicle types (e.g., Aumont et al., 2003; Kurtenbach et al., 2001; Li et al., 2011; Zhang et al., 2016).

Many field observational studies reported unknown HONO sources during the day, and various mechanisms have been proposed as efficient daytime HONO formation mechanisms. The photolysis of particle-phase $NO_3^-$ ($aNO_3^-$) < 300 nm has been previously suggested as a supplemental $NO_x$ source (Romer et al., 2018) and can be the efficient HONO production mechanism during the daytime in an aqueous environment with low pH and the presence of OH scavengers (Benedict et al., 2017a; Benedict et al., 2017b; Scharko et al., 2014; Ye et al., 2018). Another study addressed the altitudes below 300 m, where HONO deposited onto the ground surface at night and further proposed to be a significant reservoir for HONO during the day

(VandenBoer et al., 2013). Such a parameter for ground surfaces in a global model is somewhat uncertain. Moreover, the HONO source from ground surfaces may only affect the lower boundary layer while insignificantly contributing to the tropospheric HONO budget (Ye et al., 2018; Zhang et al., 2009). Furthermore, the particle-phase $NO_3^-$ photolysis can occur on both ground and aerosol surfaces ($HNO_3 + h\nu \rightarrow$ HONO) with a 2-orders-of-magnitude faster rate than its rate in the gas phase (Lee et al., 2016; Lu et al., 2018). Photolysis of ortho-nitrophenols, photoexcited $NO_2$ gas reaction ($HO_2 \times H_2O + NO_2$ $\rightarrow$ HONO), photosensitized heterogeneous conversion of $NO_2$ on grounds are all potential daytime HONO sources (Jorba et al., 2012; Lee et al., 2016; Li et al., 2014), yet the mechanisms are complicated, and their efficiency are merely evaluated for ground-based observation.

Many scholars have scrupulously addressed the effects of HONO in polluted regions as well. For instance, HONO-induced enhancements in winter daytime $HO_x$ (up to >200% for OH) and $O_3$ (6–12%) over urban sites in China have been reported (Li et al., 2011; Lu et al., 2018; Zhang et al., 2016). A box modelling study analysed the detailed budget of HONO in London and found that HONO chemistry increased OH by 20% during the day (Lee et al., 2016). A global modelling study found increments for OH and $O_3$ across the globe and throughout the troposphere, with a maximum of 30 ppb $O_3$ in Eastern Asia and slight $NO_2$ increment, although the results were evaluated with only ground-based data (Jorba et al., 2012). However, enhanced $O_3$ levels in response to additional OH production from the HONO photolysis only occur in high-$NO_x$ regions, although they can be decreased in some areas under low $NO_x$ conditions (Jorba et al., 2012). At the same time, another 3D modeling study used a constant occurrence ratio for HONO as 0.02 of $NO_x$ and reported similar patterns for $O_3$ changes regarding HONO chemistry (Elshorbany et al., 2012). The $NO_x$ reduction effects that follow the $NO_2$ conversion are suggested to be more critical over the oceans than over continental regions, with up to 20% $NO_x$ reduction and 5%–20% $HNO_3$ enhancement over ocean regions of the lower troposphere (Martin et al., 2003).

As $H_2O$ is required for the uptake of $NO_2$ on surfaces, wet surfaces have been broadly recommended as favoured surfaces for $NO_2$ uptake. Therefore, cloud droplets can be an important surface for heterogeneous reactions of $NO_2$ because they are ubiquitous in the troposphere. Heterogeneous reactions by clouds can have a similar impact as aerosol particles on tropospheric $O_3$ and OH levels (Christopher et al., 2019). However, this aspect has been overlooked many times in previous studies, leading to potential underestimating (or even dismissing) the potential effects over remote environments.

This study introduced HONO photochemical processes into the global atmospheric chemistry model CHASER-V4.0, which did not consider HONO chemistry before. The standard model configuration used basic mechanisms of HONO chemistry, while various sensitivity cases implementing other potential HONO sources were also conducted to force simulation into an agreement with the observed HONO values. The main idea for the HONO inclusion was to elaborate the model simulation for tropospheric oxidative substances while focusing on aerosol and cloud processes. The model included the detailed online calculation of $O_3$-$HO_x$-$NO_x$-$CH_4$-CO coupling and oxidation of non-methane hydrocarbons (NMHCs) (Sudo et al., 2002) and heterogeneous processes for $N_2O_5$, $HO_2$, and $RO_2$ radicals (Ha et al., 2021; Sekiya and Sudo, 2014; Sekiya et al., 2018; Sudo and Akimoto, 2007). In Section 2, we describe the approach, including the model description and configuration. In Section 3.1, simulated daytime HONO was verified with aircraft measurements for an Asian off-coast region. In addition,

our model was evaluated by the available observations for atmospheric species, including observations from aircraft, ship, ground stations, particularly addressing the roles of the HRs. Section 3.2 presents the model results for HONO distributions, verification for global effects to TCO with OMI satellite, global HONO impacts including different effects from each pathway and a discussion on the uncertainty of the calculated effects. Finally, Section 4 effectively represents the summary and concluding remarks.

## 2. Method and configurations

### 2.1. Global chemistry model

This study applied the global chemistry model CHASER (MIROC-ESM) (Sudo et al., 2002, 2007; Watanabe et al., 2011), which considered the detailed photochemistry in the troposphere and stratosphere. The chemistry component of the model, based on CHASER-V4.0, retrieved the concentrations of 94 total species and 258 chemical reactions (57 photolytic, 180 kinetic, and 21 heterogeneous reactions on tropospheric aerosol and cloud surfaces and polar stratospheric clouds) (Table 1), excluding the new HONO chemistry implemented in this study. We used the HTAP-II (Hemispheric Transport of Air Pollution) emission inventory for 2008 (https://edgar.jrc.ec.europa.eu/dataset_htap_v2, last access: 16[th], Nov 2021) for $O_3$ and aerosol precursors ($NO_x$, CO, VOCs, $SO_2$), with biomass burning emissions derived from the MACC (Monitoring Atmospheric Composition and Climate) reanalysis system (https://gmao.gsfc.nasa.gov/reanalysis/MERRA/ceop.php). The details about CHASER could be found in the earlier studies (Ha et al., 2021; Morgenstern et al., 2017; Sekiya et al., 2018). In this study, the newly added HONO system included three pathways of HONO formation and interactions: (1) gas-phase formation via the NO + OH reaction (R2), the photolysis of HONO (R1), and the reaction of HONO with OH (R3), hereafter denoted as GRs; (2) HONO direct emissions estimated from anthropogenic- and soil-$NO_x$ emissions (hereafter denoted as EM); and (3) the HONO conversion from $NO_2$ (R4, R5) and its loss on liquid/ice surfaces and aqueous aerosols (R6), which is hereafter denoted as HRs.

The investigation on heterogeneous photolysis of $HNO_3$ ($HNO_3 + h\nu \rightarrow HONO$), which was suggested as an efficient HONO source at daytime (Lee et al., 2016; Zhou et al., 2011), is presented in Chapter 3 as sensitivity cases in the effort of making simulation for daytime HONO compatible with measurement. This photolysis was simply accessed in the model via its rate using a multiply factor to the gas-phase $HNO_3$ photolysis ($HNO_3 + h\nu \rightarrow OH + NO_2$) (see Sect. 3.1.2). Another proposed daytime HONO source from the light-dependent gas-phase reaction of $HO_2$ and $NO_2$ ($HO_2 \times H_2O + NO_2 \rightarrow HONO$) (Li et al., 2014) was not investigated in this study. However, a simple gas-phase reaction of $HO_2$ and $NO_2$ ($HO_2 + NO_2 \rightarrow HONO + O_2$) (Burkholder et al., 2015), was introduced, but it did not successfully preserve the total reactive nitrogen chemistry ($NO_y$); hence, it was omitted in this study.

## 2.2. Experimental setup

The Global Emissions Initiative (GEIA) inventory (http://www.geiacenter.org/) was applied to quantify the soil $NO_x$ emissions (6 TgN yr$^{-1}$) and anthropogenic $NO_x$ emissions (45 Tg N yr$^{-1}$). Since this broadly applied inventory was not currently available for HONO, this study tentatively imposed the HONO direct emissions based on the above $NO_x$ emission inventory through a constant factor of 0.1 (10% of $NO_x$ emissions). This assumption (soils + combustion) led to a global HONO soil-emission estimate of about 0.6 TgN yr$^{-1}$, equivalent to the estimate from Porada et al. (2019), and it suggested that the anthropogenic emission for HONO is 4.5 TgN yr$^{-1}$. For HONO from exhaust sources, this factor (10%) was considerably higher than the previously reported estimate of 0.7%, derived for combustion (Xue et al., 2022b) or 0.8–2.3% for on-road vehicles (Aumont et al., 2003; Kurtenbach et al., 2001; Li et al., 2011) and 3%–6% for commercial aircraft (Lee et al., 2011). However, this factor of HONO emission (10% $NO_x$ emission) intended to show the apparent potential impacts of direct HONO sources on the tropospheric chemistry.

**Table 1: Computation packages in the chemistry-climate model "CHASER"**

| | |
|---|---|
| Base model | MIROC4.5 AGCM |
| Spatial resolution | Horizontal, T42 (2.8° × 2.8°); vertical, 36 layers (surfaces approx. 50 km) |
| Meteorology (u, v, T) | Nudged to the NCEP2 FNL reanalysis |
| Emission (anthropogenic, natural) | Industry traffic, Vegetation Ocean Biomass burning specified by MACC reanalysis |
| Aerosol | BC/OC, sea-salt, and dust BC ageing with $SO_x$/SOA production |
| Chemical process | 94 chemical species, 263 chemical reactions (gas phase, liquid phase, non-uniform $O_x$-$NO_x$-$HO_x$-$CH_4$-CO chemistry with VOCs $SO_2$, DMS oxidation (sulfate aerosol simulation) $SO_4$-$NO_3$-$NH_4$ system and nitrate formation Formation of SOA BC ageing (+) Heterogeneous reactions: 8 reactions of $N_2O_5$, $HO_2$, $RO_2$; constant uptake coefficients ($\gamma$) on types of aerosols (Ice, Liquid, Sulfate, Sea salt, Dust, OC) |

The photolysis reaction HONO + h$\nu$ → OH + NO (300 nm < $\lambda$ < 405 nm) (R1) was employed with the wavelength-dependent cross-sections following the recent study of Burkholder et al. (2015).

The kinetic of homogeneous reactions NO + OH + M → HONO + M (R2) and HONO + OH → $NO_2$ + $H_2O$ (R3) was applied with the low and high-pressure-limit rate constants, which were temperature dependent, as suggested in the aforementioned study.

In CHASER, the heterogeneous chemistry of interest was simplified as a first-order chemical loss in the aerosol phase for a species transferred from the gas phase. The rate of this pseudo loss was combined, and the first-order-loss rate for heterogeneous processes was calculated by using the Schwartz theory (Jacob, 2000; Schwartz, 1986), being simply treated with the mass transfer limitations in addition to the reactive uptake coefficient ($\gamma$) (Ha et al., 2021). Note that only surface reactions were considered in CHASER, and there was no bulk particle reaction for the HR scheme.

The uptake coefficient parameter ($\gamma$) is defined as the net probability that a molecule X undergoing a gas-kinetic collision with a surface is taken up onto the surface. An average uptake coefficient for $NO_2$ (R4) of $10^{-4}$ ($10^{-6}$–$10^{-3}$) for the conversion of aqueous aerosols and clouds has been previously suggested (Jacob, 2000; Kleffmann et al., 1998; Li et al., 2018; Lu et al., 2018). The $NO_2$ uptake by organic carbon aerosols has been reported to have similar coefficient values (Salgado-Muñoz and Rossi, 2002). The uptake coefficient for fresh black carbon is highly efficient and equals $3\times10^{-3}$ (Ammann et al., 1998; Li et al., 2018). The parameters for the uptake coefficients of R4 applied in the CHASER model are shown in Table 2.

As previous studies have noted, the fast initial uptake of $NO_2$ is observed on soot with an uptake coefficient in the range of $10^{-1}$–$10^{-4}$ (Ammann et al., 1996, 1998). However, it rapidly decreased to ~$10^{-7}$ over 5 min (Kleffmann et al., 1999) to < $4\times10^{-8}$ for 5-day aged surfaces (Saathoff et al., 2001). In organic soot, $\gamma$ is in the range of $10^{-4}$–$10^{-6}$ (Al-Abadleh et al., 2000; Ammann et al., 1996; Arens et al., 2001; Salgado-Muñoz et al., 2002). In CHASER, the $NO_2$ conversion on organic carbon and soot (R5) was tentatively applied with uptake coefficients of $10^{-4}$ and $3\times10^{-4}$, respectively, which also falls within the previously suggested range ($10^{-6}$–$10^{-3}$) considering the higher efficiency for soot (Table 2).

Also, previous laboratory experiments have introduced a wide range for the uptake coefficient of HONO by (R6), that is, $3.7\times10^{-3}$ at 178 K to $1.3\times10^{-5}$ at 200 K for the ice surface (Fenter and Rossi, 1996; Chu et al., 2000) and $4\times10^{-3}$–$4\times10^{-2}$ at 278 K (Mertes and Wahner, 1995) or 0.03-0.15 at 297 K (Bongartz et al. 1994) for liquid water surfaces. In the aerosol flow reactor experiment on deliquescent sodium chloride and ammonium sulfate droplets at 279 K, the HONO reactive uptake coefficient of 0.0028 for 85% relative humidity has been previously obtained (Harrison and Collins, 1998). In CHASER, the aforementioned reference values for HONO uptake on ice, liquid clouds, and aqueous sulfate were simply averaged to be used as a heterogeneous loss of HONO (R6) in the atmosphere (Table 2: last row).

**Table 2: Uptake coefficients for heterogeneous formation and loss of HONO**

| Reactions | | $\gamma_{ice}$ | $\gamma_{liq.}$ | $\gamma_{sulf.}$ | $\gamma_{salt}$ | $\gamma_{dust}$ | $\gamma_{oc}$ | $\gamma_{ec}$ |
|---|---|---|---|---|---|---|---|---|
| R4 | $NO_2 \rightarrow 0.5HONO + 0.5HNO_3$ | 0.0 | 0.0001 | 0.0001 | 0.0001 | 0.0001 | 0.0001 | 0.003 |
| R5 | $NO_2 \rightarrow HONO$ | 0.0 | 0.0 | 0.0 | 0.0 | 0.0 | 0.0001 | 0.0003 |
| R6 | $HONO \rightarrow NO$ | 0.002 | 0.03 | 0.003 | 0.0 | 0.0 | 0.0 | 0.0 |

### 2.3. Simulations

In this study, two main simulations, OLD and STD, and three sensitivity simulations (Table 3, no. 2–4), were conducted to isolate the distinct impacts of each pathway of the HONO chemistry for different surface types considered in the model (Table 3). The OLD simulation was run with the base model configuration without any HONO species and HONO related processes. The heterogeneous scheme in the OLD simulation contained eight reactions on $N_2O_5$ ($N_2O_5 \rightarrow 2HNO_3$), $HO_2$ ($HO_2 \rightarrow 0.5 H_2O_2 + 0.5 O_2$), and $RO_2$ ($RO_2 \rightarrow$ inert products) (Ha et al., 2021). The control case (STD) considered all three types of

HONO sources: direct emissions (EM), gas-phase reactions (GRs), and heterogeneous reactions (HRs). To quantify the effects of each mechanism using Eq. (1), two sensitivity cases (GR, GR+HR) intentionally implemented GRs (R1, R2, R3) into the OLD case and HRs (R1, R2, R3, R4, R5, R6) into the GR case, respectively. GR+HR (cld) was another sensitive case like GR+HR, with HRs on aerosols excluded to investigate the different effects of clouds and aerosols. Eq. (1) determines the effects of each mechanism on atmospheric species $i$ ($i$ = OH, $O_3$, $NO_x$, CO) by concentration differences of $i$ in two relevant

cases being compared to that in the OLD case.

$$E_i = \frac{(Case1_i - Case2_i)}{OLD_i} * 100 \ (\%), \tag{1}$$

where $Case1_i$ and $Case2_i$ are the concentrations of $i$ in two separate cases: GR and OLD cases for the pure effects by the gaseous mechanism; GR+HR and GR cases for the effects of heterogeneous mechanisms; STD and GR+HR cases for the HONO emissions effects; and GR+HR(cld) and GR cases for the effects of heterogeneous reactions that exclusively occur on

ice and cloud particles.

**Table 3: Sensitivity simulations in this work.**

| No. | Simulation ID | HRs (HONO) | | GRs (HONO) | EM (HONO) |
|-----|---------------|--------|---------|------------|-----------|
| | | clouds | aerosols | | |
| 1 | OLD | | | | |
| 2 | GR | | | × | |
| 3 | GR+HR(cld) | × | | × | |
| 4 | GR+HR | × | × | × | |
| 5 | STD | × | × | × | × |

### 2.4. Observation data for model evaluation

We evaluated the OLD, STD, and sensitivity simulations with aircraft, ship-based, ground-based, and satellite measurements. The observational information and locations of the ship/aircraft tracks and surface sites for the observations used in this study

are summarized in Table 4, Figure 1, Figure 6, and Figure S6.

Daytime HONO concentrations were analyzed by using the DLR-HALO aircraft (Operator Deutsches Zentrum für Luft- und Raumfahrt - High-Altitude and Long-Range research aircraft) measurements made during the EMeRGe-Asia (Effect of Megacities on the Transport and Transformation of Pollutants on the Regional to Global scales) campaign in March and April

2018, over an off-coast region between Korea (including the Jeju Island as the part of the domain), Taiwan, and the Philippines (http://www.iup.uni-bremen.de/emerge/home/). The measuring time falls in the range of 0:00 UTC to 9:00 UTC, around 8:00 to 17:00 in local time (UTC+8). The payload during EMeRGe-Asia mission could be retrieved from the similar mission EMeRGe-Europe (Andrés Hernández et al., 2021). Verification with EMeRGe data helps explore the daytime HONO chemistry mechanisms in the free troposphere.

To verify the vertical profiles of atmospheric species for the oceanic tropospheric environment, ATom1 aircraft measurements (https://espo.nasa.gov/atom/content/ATom) for $NO_2$, OH, CO, and $O_3$ during August 2018 were employed. We also utilized the ship-based observational data from the *R/V Mirai* cruise (http://www.jamstec.go.jp/e/about/ equipment/ships/mirai.html) undertaken by Japan Agency for Marine-Earth Science and Technology (JAMSTEC) for surface CO and $O_3$ in summers 2015-2017 along the Japan-Alaska routes. The monthly data from 45 stations during 2010–2016 were used to verify aerosol surface concentrations (sulfate, nitrate) and trace gases ($HNO_3$, $NO_x$, $O_3$) in the Acid Deposition Monitoring Network in eastern Asia (EANET: https://www.eanet.asia/). We also used the European Monitoring and Evaluation Programme (EMEP: https://www.emep.int/) data, which compiles observations over 245 European stations. To this end, simulated tropospheric column ozone was also evaluated by using tropospheric column $O_3$ (TCO) derived from the OMI (Ozone Monitoring Instrument) spaceborne observations (https://daac.gsfc.nasa.gov/). For these evaluations and verifications, the model data were compiled in the monthly or hourly timestep, interpolated corresponding to the observed data time step and coordinates.

A model bias for each species was calculated as the difference between the simulated and observed concentrations, as shown in Eq. (2), where N is the total number of data points used in the calculation.

$$bias = \frac{\sum_1^N Model - Observation}{N} \tag{2}$$

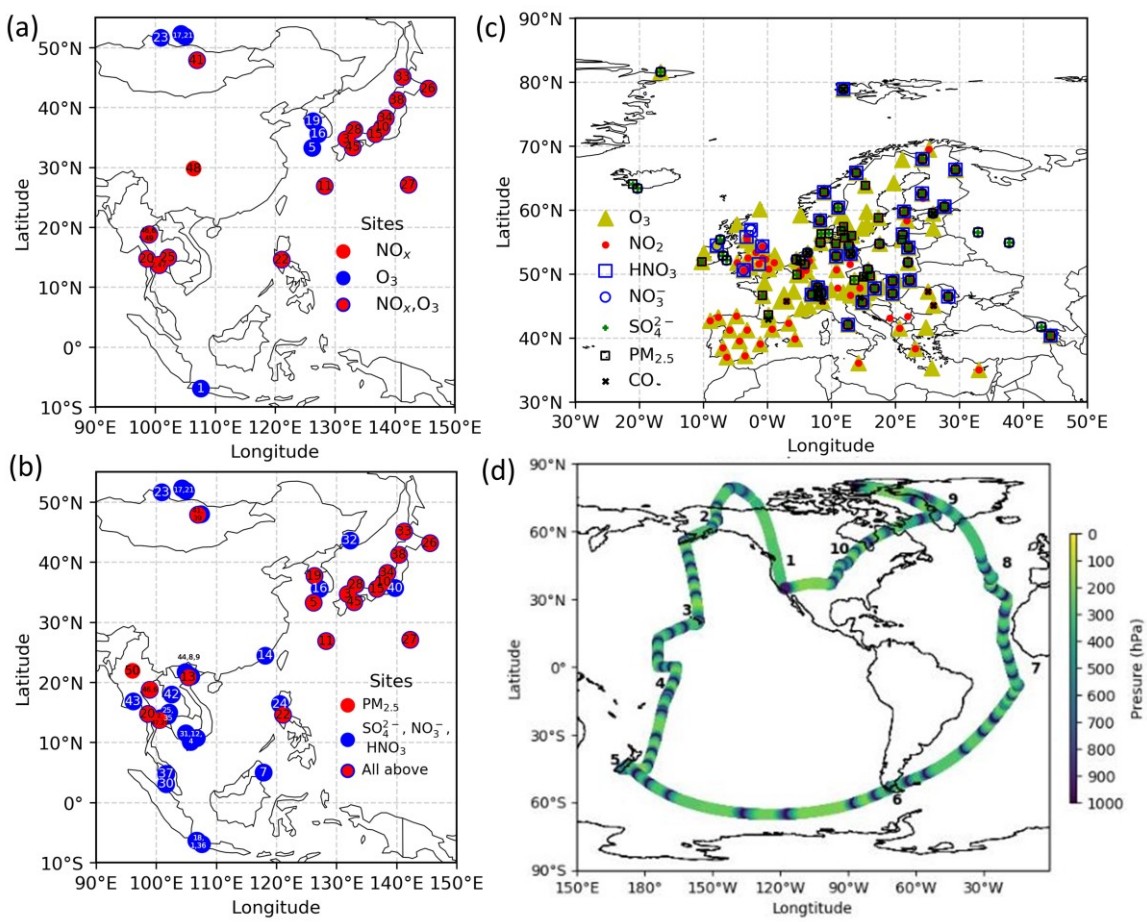

Figure 1: Location of measurements. (a) EANET stations for NOx and O3 and (b) for PM2.5, SO4^2, NO3^-, and HNO3, (c) EMEP stations, and (d) ATom1 cruising altitudes are plotted. In (a) and (b), each number described station name (see Table S1). In (d), numbers show flight tracks.

Table 4: Lists of the datasets used in this study for verification. Related simulations with their original model timestep are interpolated to the comparing timestep.

| Verified species | Regions | Dataset name | Time | Measuring step | Model step | Interpolating step |
|---|---|---|---|---|---|---|
| PM2.5, SO4^2-, NO3^-, NOx, O3, HNO3 | East Asia | EANET (station) | 2010–2016 | Daily to 2-weekly | Daily | Monthly |
| PM2.5, SO4^2-, NO3^-, NOx, O3, CO | Europe | EMEP (station) | 2010–2016 | Hourly | Daily | Monthly |
| CO, O3 | Australia – Indonesia – Japan – Alaska | *Mirai* (vessel) | 8,9/2015 1,8,9/2016 7,8,9/2017 | 30 min | 1 h | 30 min |
| NO2, OH, CO, O3 | Pacific, Atlantic ocean, Greenland, North America | ATom1 (aircraft) | 8/2016 | 30 min | 1 h | 30 min |

| Tropospheric column ozone (TCO) | 60°S - 60°N | OMI (Satellite) | 2010–2016 | Daily | Daily | Monthly |
|---|---|---|---|---|---|---|
| HONO | Jeju-Korea, Taiwan, and Philippine | EMeRGe (aircraft) | 17/3/2018–4/4/2018 | 15–30 s | Hourly | 14–30 s |

## 3. Results and Discussion

### 3.1. Verification and validation of model simulations for cloud fraction, surface area density, atmospheric species, and effects on HONO mechanisms

#### 3.1.1. Cloud fraction and surface area density for cloud and aerosols

In this study, besides $NO_2$ conversion onto clouds and aqueous particles (R4), the losses of HONO onto the ice and liquid clouds (R6) are also included. Therefore, for accurate simulations of HRs, we need to examine the cloud distribution. The CHASER model applied the common cloud maximum-random overlap assumptions (MRAN) in the radiation and cloud microphysics schemes as other general circulation models to estimate the distribution of the cloud fraction. The verification by using the satellite observation data ISCCP D2, CALIPSO-GOCCP, and reanalysis data JRA55 generally revealed good correlation, whereas notable (10–20%) underestimation for the entire troposphere was yet salient. During June – July – August (JJA), CHASER's cloud fraction was likely overestimated for the lower troposphere of the North Pacific (NP) region (10–20% compared to JRA55 reanalysis data). This finding indicated that thorough scrutiny of any impacts in this region is highly required (see the discussion in Section 3.2). Note that more detailed information for cloud verification for CHASER has been provided by Ha et al. (2021).

The heterogeneous processes by clouds and aerosol particles were parameterized by using surface area density (SAD) estimations alongside the cloud fraction and aerosol concentration. During DJF, the simulated total SAD was attributed to all types of aerosols. However, for JJA, liquid clouds and sulfate aerosols were the principal SAD sources. This was a peculiarly visible pattern for the northern polar and mid-latitude maritime regions. The performance for aerosol SAD in our model was in line with the earlier report by Thornton et al. (2008), except for sea salt density, which was very low in our model (up to 2 $\mu m^2$ $cm^{-3}$) compared to their work (up to 75 $\mu m^2$ $cm^{-3}$). This disagreement might be ascribed to the two models' different size distributions for sea salt. The calculated SAD for the liquid cloud was two orders of magnitude higher than SAD for ice cloud and total aerosols. Liquid cloud SAD maximized at ~ 800 hPa in the tropical convective systems and over the midlatitude storm tracks, reaching ~ 50,000 $\mu m^2$ $cm^{-3}$ at the surface of the North Pacific region in JJA. Sulfate aerosols dominated above 600 hPa for the Northern Hemisphere (~ 20 $\mu m^2$ $cm^{-3}$) among the total aerosol surface area, followed by organic carbons and soil dust (~ 10 $\mu m^2$ $cm^{-3}$ in JJA). At the surface layer, sulfate aerosols were prevalent in DJF for the Chinese region (> 1,000 $\mu m^2$ $cm^{-3}$), north-eastern U.S. (~ 500 $\mu m^2$ $cm^{-3}$), and North Pacific region in JJA (~ 250 $\mu m^2$ $cm^{-3}$). SAD for soil dust dominated in desert regions, with annual average values > 100 $\mu m^2$ $cm^{-3}$. Organic carbon (OC) was dominant in winter over

biomass burning regions such as China (up to 1,000 μm$^2$ cm$^{-3}$) and South Africa (up to 800 μm$^2$ cm$^{-3}$). For the Chinese region, SAD for black carbon (BC) could reach 600 μm$^2$ cm$^{-3}$ in DJF and 75 μm$^2$ cm$^{-3}$ in India. The total-aerosol SAD for the northern high-latitude and mid-latitude oceans was ~ 75 μm$^2$ cm$^{-3}$, consistent with the estimation by Thornton et al. (2008).

### 3.1.2. Daytime concentrations of HONO and other atmospheric species

This section evaluated CHASER-based HONO estimates using the HONO measurements collected during the EMeRGe campaign off-coast eastern Asia in spring 2018 (Andrés Hernández et al., 2021). This is the first global HONO modelling work using EMeRGe as the validation source. The HONO measurements in the free troposphere could provide essential information on the underlying gas-phase and heterogeneous HONO formation mechanisms as most current HONO measurements were conducted in the surface air. The daytime HONO concentration was retrieved from the aircraft-borne limb measurements using the HALO mini-DOAS (differential optical absorption spectroscopy) instrument, in which the absorbed UV light (310–440 nm) by HONO was detected (Hüneke et al., 2017). The mini-DOAS's measurement method relies on near-UV/VIS/IR skylight spectroscopy in nadir and limb geometry. Data evaluation consists of three steps: (1) retrieval of slant column densities (SCDs) of trace gases by the DOAS method (Platt and Stutz, 2008), (2) forward radiative transfer modeling for each measurement using McArtim (Deutschmann et al., 2011); and (3) retrieval of concentration through a new scaling method for UV/VIS data (Stutz et al., 2017; Hüneke et al., 2017; Werner et al., 2017; Kluge et al., 2020; Rotermund et al., 2021).

Additional sensitivity runs were conducted to explore potential HONO sources during the daytime (Table 5). The ratR4+CLD case is run in an attempt to produce more HONO from heterogeneous sources by altering the HONO: HNO$_3$ yield ratio in (R4) to 0.9: 0.1, and $\gamma_{liq.}$ increased a hundred-fold ($10^{-4}$ → $10^{-2}$). The main idea here is to evaluate whether the missing HONO source was sensitive to cloud uptake in this region or not. The maxST case maximized the uptake coefficients (γ-values) of NO$_2$ on organic and black carbons to 0.1 (R4, R5), to estimate the separate role of soot uptake under daytime conditions (George et al., 2005; Monge et al., 2010; Ndour et al., 2008), which could achieve an unrealistically high γ-value of $10^{-1}$ (Ammann et al., 1998; Kalberer et al. 1999). In three other runs (JANO3-A, JANO3-B, JANO3-C), the photolysis of aerosol nitrate / adsorbed HNO$_3$ on the ground and other surfaces (NO$_3^-$/HNO$_3$) were examined, simply as HNO$_3$ + hν → HONO (R7)). These heterogeneous photolyses of HNO$_3$ were previously proposed as potential HONO sources at day (Lee et al., 2016; Scharko et al., 2014; Zhou et al., 2011). Because aerosol nitrate and aqueous surfaces are ubiquitous in the atmosphere, the photolysis (R7) was simply set for the gaseous HNO$_3$ species to occur in particular model spatial grids exposing ground surfaces and sufficient surface area density for aerosols and clouds. The photolysis (R7) was taken at a faster rate by two orders of magnitude than the gas-phase photolysis rate of HNO$_3$ (HNO$_3$ + hν → OH + NO$_2$) (Zhou et al., 2011) and presumably yield 100% HONO to access the maximum effects by this photolysis (Lee et al., 2016). This setting allows (R7) not only to occur at the surfaces of particles but also in the gas and bulk phases. However, in this test, (R7) generally refers to surface-catalyzed photolysis or heterogeneous photolysis of HNO$_3$. The JANO3-A case investigated the photolysis

of adsorbed HNO$_3$ on ground surfaces by implementing (R7) for the first vertical layer (z=1). The JANO3-B explored photolysis of nitrate particles and adsorbed HNO$_3$ gas on both grounds and aerosol surfaces, applying (R7) for model grid cells with the SAD of $10^{-6} \sim 10^{-4}$ cm$^2$ cm$^{-3}$ (100 to 10,000 µm$^2$ cm$^{-3}$) to use $10^{-4}$ cm$^2$ cm$^{-3}$ threshold to exclude cloud surfaces (Sect. 3.1.1). The JANO3-C case examined (R7) for regions present of all particles with SAD $\geq 10^{-7}$ cm$^2$ cm$^{-3}$ (10 µm$^2$/cm$^3$). The SAD of $10^{-6}$ and $10^{-7}$ cm$^2$ cm$^{-3}$ was supposed to be the threshold for continental aerosols. The maxST and JANO3-B / JANO3-C cases were also combined in two additional cases (maxST+JANO3-B and maxST+JANO3-C, respectively), given the contrary effects on NO$_2$-O$_3$-CO chemistry of these cases might be neutralized, is discussed next. Other tests examined the possible HONO sources from aviation crafts (AIRC), amplified emissions (EMx8), amplified homogeneous HONO formation (R2) (GRx8), which descriptions were listed in the supplement Table S3.

The correlation coefficient (R) and model biases against EMeRGe for HONO were shown in Table S4. As seen for the STD run, general underestimations of HONO simulations was identified, in which better correlations were found at 1000–2000 m (R=0.31–0.49). Vertical profiles for HONO and other species (NO$_2$, O$_3$, CO), retrieved from the EMeRGe flights, were applied for the measurement-based model evaluation (Figure 2). The model discrepancies for the measurement for HONO ($\Delta_{HONO}$) and NO$_2$ ($\Delta_{NO2}$) in each flight trajectory, i.e., from Taiwan to South Korea, Japan, and the Philippines, were separated into bins of altitude ranges 0-1,000-3,000-5,000-6,000 m (Figure 3). The frequency distributions of $\Delta_{HONO}$, $\Delta_{NO2}$, $\Delta_{O3}$, and $\Delta_{CO}$ were shown in Figures S7 and S8.

Figure 2(a) shows the vertical average score (cruising altitudes ± 500 m) for the measured (blacks) and simulated HONO concentrations in STD (reds) and those results of sensitivity cases. The measured daytime HONO concentration was close to the boundary layer (below 1,000 m) over Taiwan, averaged at 115 ppt, and was peaked at ~250 ppt. Also, the HONO concentration decreased up to 9,000 m (± 500 m), with mean values dropping from 70 ppt (2,000 m ± 500 m) to < 20 ppt (5,000 m ± 500 m) and < 10 ppt above. These measured HONO values for this Asian coastal region were surprisingly high, which range from 10-115 ppt for 2,000 m ± 500 m altitudes, compared to Wang's report of < 100 ppt (maximum) and < 30 ppt (4 daytime hours means) for 1,500-2,000 m altitudes measured by a MAX-DOAS at a station nearby the HONO source (Wang et al., 2019). This indicates that the source of HONO during EMeRGe might relate to other mechanisms than emission sources. In this study, the simulated HONO concentration in the STD case significantly underestimated the observations. They reached only 30–70 ppt at 1,000 m and nearly zero from 2,000 m upward (Figure 2(a): red versus black triangles for the simulation and the observations, respectively). These discrepancies indicate a significant unknown HONO source during the daytime, although the proposed heterogeneous HONO formation mechanisms were incorporated in our model. This finding adds another instance of evidence about missing HONO sources in the polluted boundary layer and free troposphere (e.g., Kleffmann et al., 2003; Li et al., 2014; VandenBoer et al., 2013; Xue et al., 2022a; Ye et al., 2018).

**Table 5: Additional sensitivity simulations in this work.**

| No. | Simulation ID | Description | Note |
|-----|--------------|-------------|------|
| 1 | maxST | $\gamma_{oc}$ and $\gamma_{ec}$(R4, R5) = 0.1 | See Table 2 for $\gamma$-values in STD |
| 2 | ratR4 | NO$_2$ $\rightarrow$ 0.9HONO + 0.1HNO$_3$ (R4) | Product ratio is 0.5:0.5 in STD |

| 3 | ratR4+CLD | ratR4 and $\gamma_{liq.}$(R4) = 0.01 | = 0.0001 in STD |
| 4 | JANO3-A | Add HNO$_3$ + hν → HONO (R7) (z=1, rate = 100 × rate of HNO$_3$ + hν → OH + NO$_2$) | HONO from HNO$_3$ photolysis (adsorbed on ground surfaces) (Lee et al., 2016) |
| 5 | JANO3-B | Add (R7) (100 < SAD < 10,000 μm$^2$cm$^{-3}$) | HONO from HNO$_3$ photolysis (adsorbed on ground and aerosol surfaces for continental regions excluding cloud surface) |
| 6 | JANO3-C | Add (R7) (SAD ≥ 10 μm$^2$cm$^{-3}$) | Similar to JANO3-B but using a larger SAD threshold |
| 7 | maxST+ JANO3-B | $\gamma_{oc}$ and $\gamma_{ec}$(R4, R5) = 0.1 Add (R7) (100 < SAD < 10,000 μm$^2$cm$^{-3}$) | Combination of maxST and JANO3-B cases |
| 8 | maxST+ JANO3-C | $\gamma_{oc}$ and $\gamma_{ec}$(R4, R5) = 0.1 Add (R7) (SAD ≥ 10 μm$^2$cm$^{-3}$) | Combination of maxST and JANO3-C cases |

In Figure 3, which shows model discrepancies, the measured NO$_2$ below 3,000 m (±500 m) close to land was well captured in the model (Figure 3(a,e,i): magentas and greens), with 34% of the data being quite close for NO$_2$ (±70 ppt) (Figure S7(d)). However, the modelling still underestimated the simulated HONO mixing ratio by up to 250 ppt (Figure 3(a): green). Over the off-coast of Taiwan bound to Japan, NO$_2$ was overestimated by up to 600 ppt in the model, corresponding to 20–70 ppt missing HONO (Figure 3(a,e): small oranges left of vertical line). The missing HONO can be driven by low HONO emission from land and low uptake of NO$_2$ on organic carbon and soot, as the amplified EMx8 and maxST cases could alleviate the model underestimates for HONO (Figure 3(a vs b): oranges, (a vs c and e vs f): greens and magentas). The model also underestimated O$_3$ and CO, usually by 25 ppb O$_3$ (freq. 79%) and 100 ppb CO (freq. 60%) (Figures S10, S11), which were larger than the model biases against ATom1 observations (Sect. 3.1.3; Figure 5) because of possible inland influence. More accurate and detailed emission inventory for substances such as HONO, NO$_x$ and CO was thus sensible as this region is the outflow of the Pearl River Delta and Yangtze River Delta regions. Besides the uptake on organic and black carbon, identified in the maxST simulations, we identified the NO$_2$ uptake on sulfate nearly as important through a parallel test (not shown). Especially, the heterogeneous photolysis of HNO$_3$ couldn't provide a significant HONO amount near Taiwan, South-Korea, Japan (in JANO3-B and JANO3-C cases) and a small HONO amount for the Philippines-bounding route (in JANO3-C case) for the altitudes below 2,500 m (Figure 3(d,h): greens, oranges, blues).

For the middle troposphere (5,000–6,000 ± 500 m) over the Taiwan Island, too abundant NO$_2$ was predicted by the model during the cruises bounding to South Korea (up to 40 ppt) and the Philippines (up to 20 ppt) (Figure 3(m,p): small greens and blues left of the vertical line). These overabundances might hint on the deep stratospheric intrusion in springtime that caused imperfect downward mixing fluxes (Lin et al., 2012; Stohl et al., 2003; Trickl et al., 2014). This excessive NO$_2$ and the corresponding missing HONO were also sensitive to the AIRC and GRx8 cases (Figure 3(n,q)), indicating that aircraft-exhaust of HONO could adjust the HONO/NO$_2$ ratio and more homogeneous HONO production might contribute more, given the high abundances of oxidizing substances at these altitudes. The possibility of emission of aviation-induced particles on which NO$_x$ to HONO conversion could reach 45% (Meilinger et al., 2005) could support the need for NO$_2$ reduction and HONO formation for this height across EMeRGe's near-land domains. Moreover, the surface-catalyzed photolysis of HNO$_3$ in the JANO3-C runs could serve as an efficient source and greatly reduced the model negative bias for HONO at 6,000 m. However, the NO$_2$

reproduction in this case was too strong (because less $NO_x$ is removed via $HNO_3$ deposition), which turned the model bias for $NO_2$ become overestimates (Figure 3(s)).

The model underestimation for HONO was also associated with the concurrent underestimation of $NO_2$, observed more often at the altitudes of > 1000 m. The likely erroneous $NO_2$ concentrations of ~1.8 ppb (1,000 m) and ~220 ppt (3,000 m) across Taiwan, linked with lacked HONO of as high as 290 ppt (1,000 m)  and 140 ppt (3,000 m) (Figure 3(e,i)). These likely inadequate $NO_2$ abundances could be partially alleviated through the enhanced $HONO/NO_x$ emission ratio and more efficient $NO_x$ recycling process in the ratR4+CLD cases, respectively (Figure 3(g,j)). Here, the missing HONO was largely supplemented only at ~1,000 m over the marine environment (Taiwan-Japan cruise), when we identified more products for HONO on cloud in the ratR4+CLD case. (Figure 3(e,g): red-orange diamonds). At ~6,000 m, small deficits of 60 ppt $NO_2$ corresponding to ~10 ppt HONO (Figure 3(p): oranges and magentas) might correspond to lightning $NO_x$ emissions (Sudo et al., 2002) and stratospheric sources. Some homogeneous mechanisms at ~6,000 m as in the GRx8 could be effective (Figure 3(n,o)). Moreover, the heterogeneous photolysis of $HNO_3$ in the JANO3-C case could be an effective HONO supplement above 5,500 m (Figure 3(s)), while this photolysis acted as a $NO_2$ production mechanism at any altitudes.

In general, the upper limit for the aerosol uptake coefficients (maxST case) may be applicable for the lowest cruising altitudes, which induced the increase of modelled HONO levels during both daytime and nighttime (Figure S10). The photolysis of adsorbed $HNO_3$ on ground surfaces implemented in the JANO3-A case was impractical to be a source for HONO during EMeRGe, as this case only provided a mild HONO amount at a thin surface layer (<500 m; not shown). Fortunately, the surface-catalyzed photolysis of $HNO_3$ in JANO3-B and JANO3-C cases could remedy the model-measurement discrepancies, i.e. $R_{HONO}$>0.6 and the model bias for HONO was reduced from -112 ppt (STD) to -22 ppt (JANO3-B) and -18 ppt (JANO3-C) for 0-500 m altitudes (Table S4). The HONO source from this photolysis of $HNO_3$ was sufficient for continental and near-land regions. In particular, the photolysis of $HNO_3$ adsorbed on particles with smaller SAD (JANO3-C case) was responsible for the 500 – 3,000 m atmosphere around Philipines and South Korea and at higher altitudes where robust solar radiance might enhance the $HNO_3$ photolysis. In the combined cases (maxST+JANO3-B and maxST+JANO3-C), HONO production was boosted, and the estimated $NO_2/O_3$ concentrations were best captured for 2,000 – 5,000 m (±500 m) altitudes (Figure 2(a,b,d): black diamonds vs orange circles). Furthermore, the sensitivity cases including combined cases changed the global tropospheric effects differently, as discussed in Sect. 3.2.3.

The remaining drawbacks in reproducing HONO and other atmospheric species ($NO_2$, $O_3$, CO) by model urges further elucidation of efficient HONO formation mechanisms. To this end, one needs (1) to elaborate the combined HONO production mechanisms from enhanced $NO_2$ aerosol-uptakes and $HNO_3$ photolysis alongside testification other potential HONO formation mechanisms and $NO_x$ recycling processes, (2) to simulate the lower and upper limits for the uptake coefficients of $NO_2$ on aerosols and clouds, (3) to provide better emission inventories for anthropogenic sources of pollutants from Southeast Asia and East Asia, lightning-produced $NO_x$ and $HO_x$, and aviation-induced aerosols, and (4) to improve the vertical mixing and air mass transport from the stratosphere.

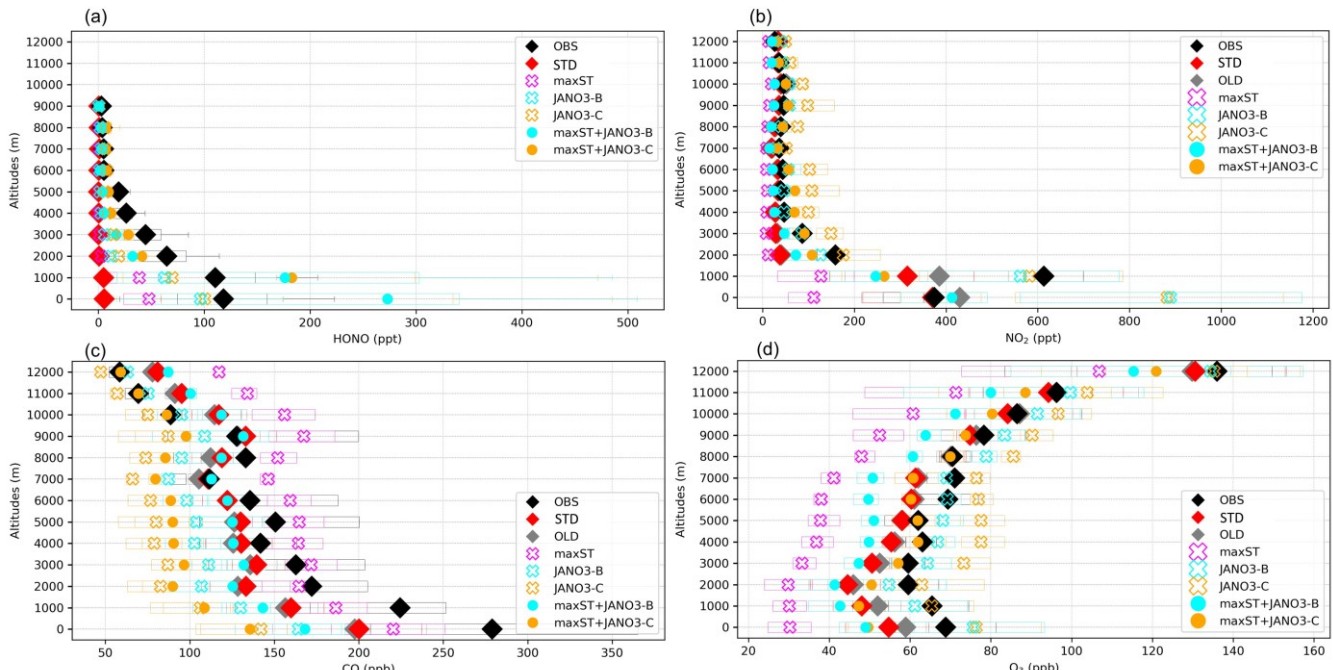

Figure 2: Vertical profiles of (a) HONO, (b) NO₂, (c) CO, and (d) O₃ measured in EMeRGe campaign and calculated in the sensitivity runs. Diamonds (for OBS, STD, OLD), batches (for maxST, JANO3-B, JANO3-C cases), and filled circles (for maxST+JANO3-B, maxST+JANO3-C cases) show mean vertical concentrations, and the corresponding boxes indicate $25^{th}$ - $75^{th}$ value ranges. In panel (a), whiskers with two caps show min and max HONO levels; all sensitivity runs are shown except OLD (the case without HONO chemistry). In all plots, black is for observation (OBS), colours are for simulations: STD (red), OLD (grey), maxST (magenta), JANO3-B (cyan), JANO3-C (orange), maxST+JANO3-B (filled cyan), and maxST+JANO3-C (filled orange).

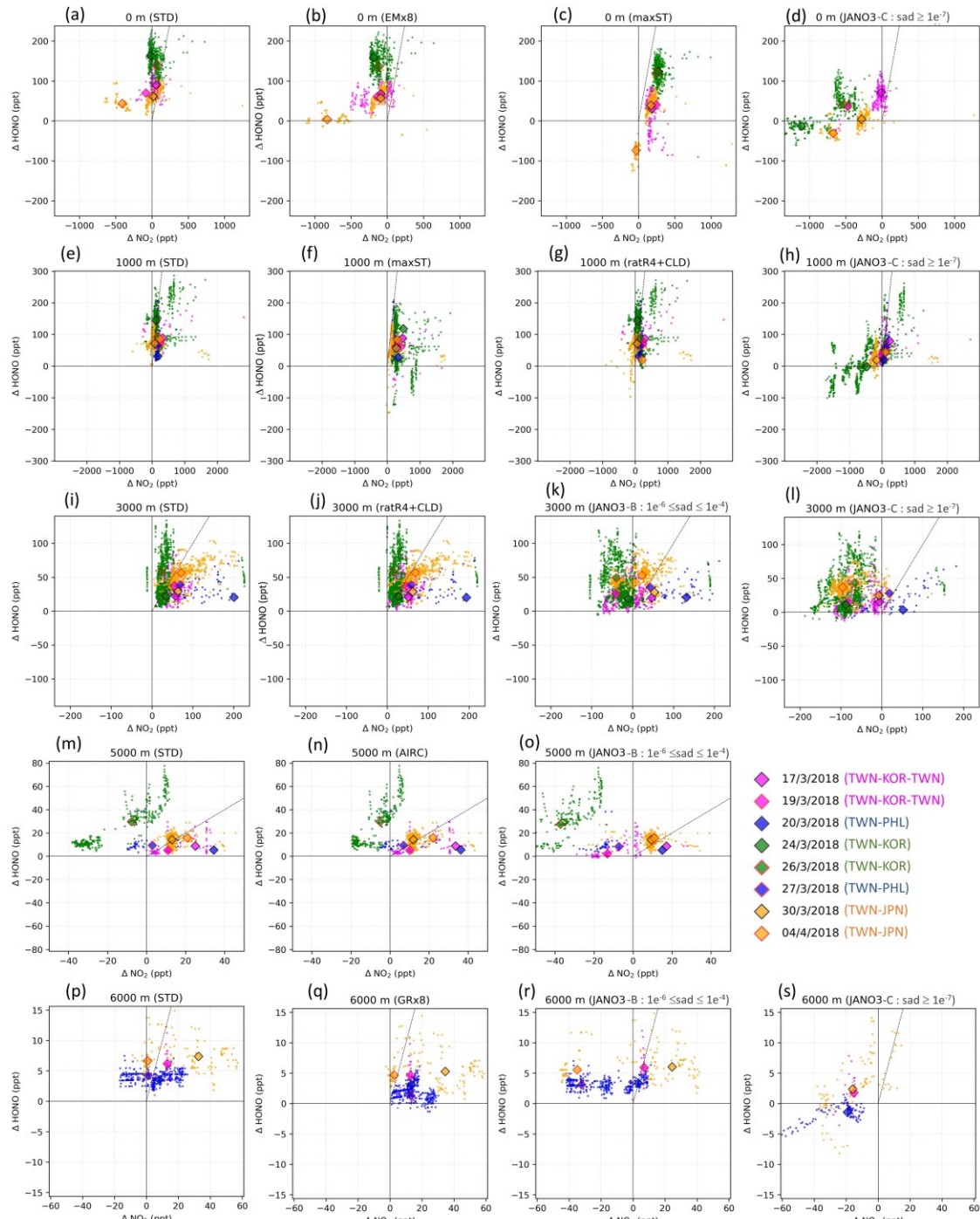

**Figure 3: Model's discrepancies from measurements for HONO ($\Delta_{HONO}$) versus that for NO₂ ($\Delta_{NO2}$). Only results from STD (first column) and helpful sensitivity cases (second, third, and fourth columns) are plotted. The scale is shared for each row. The altitude range (0, 1000, 3000, 5000, 6000 m ± 500 m) and the sensitivity case names are shown at the top of each panel. Small points represent discrepancies distribution (observation − model). Diamonds mark the median point of each cruise distribution. Edge and fill colours indicate flight cruises (see legend). Vertical, horizontal, diagonal lines show $\Delta_{NO2} = 0$, $\Delta_{HONO} = 0$, $\Delta_{NO2} = \Delta_{HONO}$, respectively.**

### 3.1.3. $NO_2$, OH, $HO_2$, $O_3$, and CO concentrations within the oceanic free troposphere

The model performance of the free troposphere was evaluated through the atmospheric tomography (ATom1) aviation in August 2016 for $NO_2$, OH, CO, $HO_2$, and $O_3$. The STD run reconstructed the chemical field observed in ATom1 with moderate or strong positive correlations for $NO_2$, OH, CO, and $O_3$ ($R_{NO2}$=0.730, $R_{O3}$=0.751, $R_{OH}$=0.579, $R_{CO}$=0.659; Table S2). For the NP region, the model correlations for these species were slightly lower ($R_{NO2}$=0.621, $R_{O3}$=0.609, $R_{OH}$=0.407, $R_{CO}$=0.596; Table S2). The $R$ values for $NO_2$ and CO were consistently higher in the STD run than those in the OLD run, while for OH and $O_3$, the $R$ values are only improved for the NP region (Table S2).

Figure 4 shows measured (grey) and simulated (red and black) $NO_2$, $O_3$, OH, CO concentrations and the effects of including HONO in the simulation for the NP region (flight #2 on August 3rd). Figure 5 displays vertical profiles of the model biases in STD vs OLD cases and photochemical effects by each HONO formation mechanism. Here, the data in all flights or in the NP region were classified based on the air pressure from 1000–200 hPa (±50 hPa) and separated into nine bins. In the NP region, the OLD run (black lines) tended to overestimate $NO_2$, $O_3$, OH, and $HO_2$ but it underestimated CO at the lower troposphere, whereas the unsteady discrepancies at the upper layer were visible (Figure 4(a,b,e,f)). All five species tended to be underestimated near the tropopause (300–400 hPa) and to be overestimated in the lower stratosphere (Figure 5(e-h,u)). The HONO inclusion in STD run (red lines) reduced $NO_2$, OH, $O_3$ and increased CO levels, thereby, dwindling the model biases for $NO_2$, OH, and CO in the NP region except near the tropopause (Figure 5(e,g,h)). $HO_2$ was reduced near the surface layer and increased from the middle troposphere (Figure 5(v,w)), reducing model bias for most parts of $HO_2$'s vertical profile (Figure 5(t,u)). The reduction in $HO_2$ level near the surface might follow similar cloud-effect as that for OH, which turned to minor increases in $HO_2$ level at middle and high atmosphere, given lower $HO_2$ level at these altitudes.

In the NP region, the surface $NO_2$ level was reduced under the effects of HONO uptake on clouds (Figure 4(c): orange bars in vertical pink shades and Figure 5(p)). Hence, $O_3$ and OH were correspondingly reduced as their formations are asumably limited in the absence of sufficient $NO_x$, that is, lacking atomic oxygen from $NO_2$ photolysis and OH formation via $HO_2$ + NO $\rightarrow$ OH + $NO_2$ (Figure 4(d,g): orange bars in pink columns and Figure 5(q,r)). Near the surface, aerosol HRs only slightly affected atmospheric species, whereas at high altitudes, the aerosol uptake was more relevant, especially for $O_3$ concentrations (Figure 4(c,d,g,h) and Figure 5(q-s): green bars), due to the contribution of aerosol direct effect to $O_3$ level (Xing et al., 2017). The dominant cloud effects near the surface appeared plausible for an ocean region with high cloud fractions at the lowest layer (Figure S2). GRs also affected OH, $O_3$, $HO_2$, and CO, whereas the effects manifest in the upper troposphere rather than in the lower troposphere. This was likely the most influential factor that increases OH, $HO_2$, and $O_3$ levels at these high altitudes (Figure 4(d,g) and Figure 5: blue bars). The additional HONO from direct emissions had minor effects on $NO_2$ and OH but contributed to the reductions of $O_3$ and CO at high altitudes (Figure 4(c,d,g) and Figure 5(q,s): red bars). At 900 hPa, the HONO emissions significantly reduced $NO_2$ near the continental areas (Figure 5(l): red bars) due to its uptake by particles. These effects of the HONO chemistry in the STD simulation somewhat reduce the model biases for $NO_2$, $O_3$, OH, $HO_2$, and CO (Figure 5(a-h,t,u): red numerical texts are the percentage reduction in model bias). Note that these biases were very

pronounced near-surface (~1000 hPa) in the NP region (51.7% for $NO_2$ and 77.3% for OH). To capture the patterns identified by observations in the upper troposphere, except the NP region, more robust increases for $NO_2$, OH, and $HO_2$ levels were still required (Figure 5(a,c,t)). At these altitudes, $NO_x$ and $HO_x$ sources from lightning (Brune et al., 2021) or aviation could also be relevant, as discussed in Section 3.1.2.

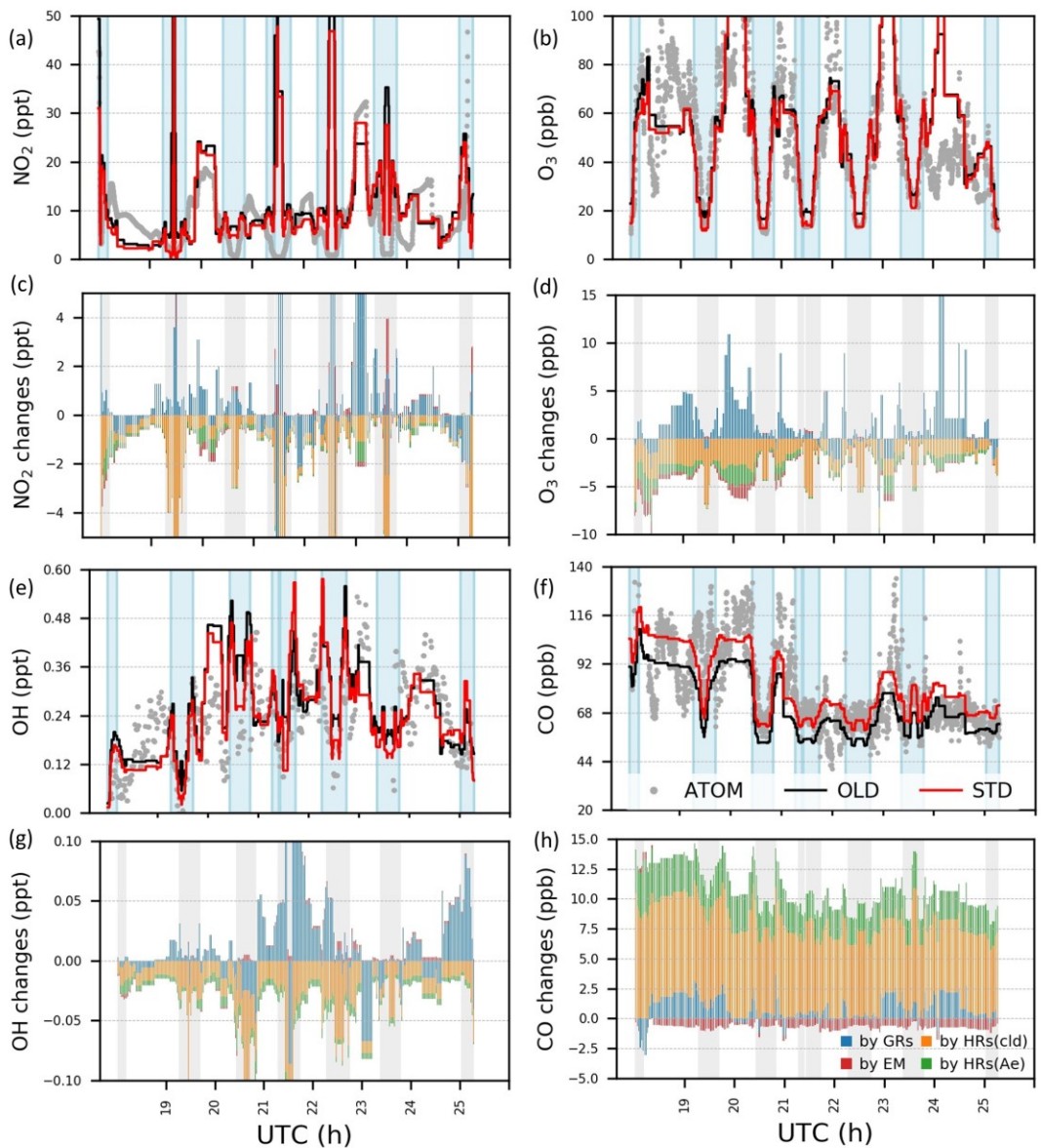

**Figure 4: Concentrations and variations by HONO chemistry for $NO_2$, $O_3$, OH and CO during ATom1 flight #2 (198–210° E, 20–62° N). In (a-b, e-f), concentrations by observation (grey dots), simulations in OLD case (black lines), and in STD case (red lines) are plotted. In (c-d, g-h), changes in concentrations by GRs (blue bars), EM (red), HRs on clouds (orange), and HRs on aerosols (green) are plotted. Vertical blue and grey columns reflect the data for the regions with air pressure $P > 500$ hPa.**

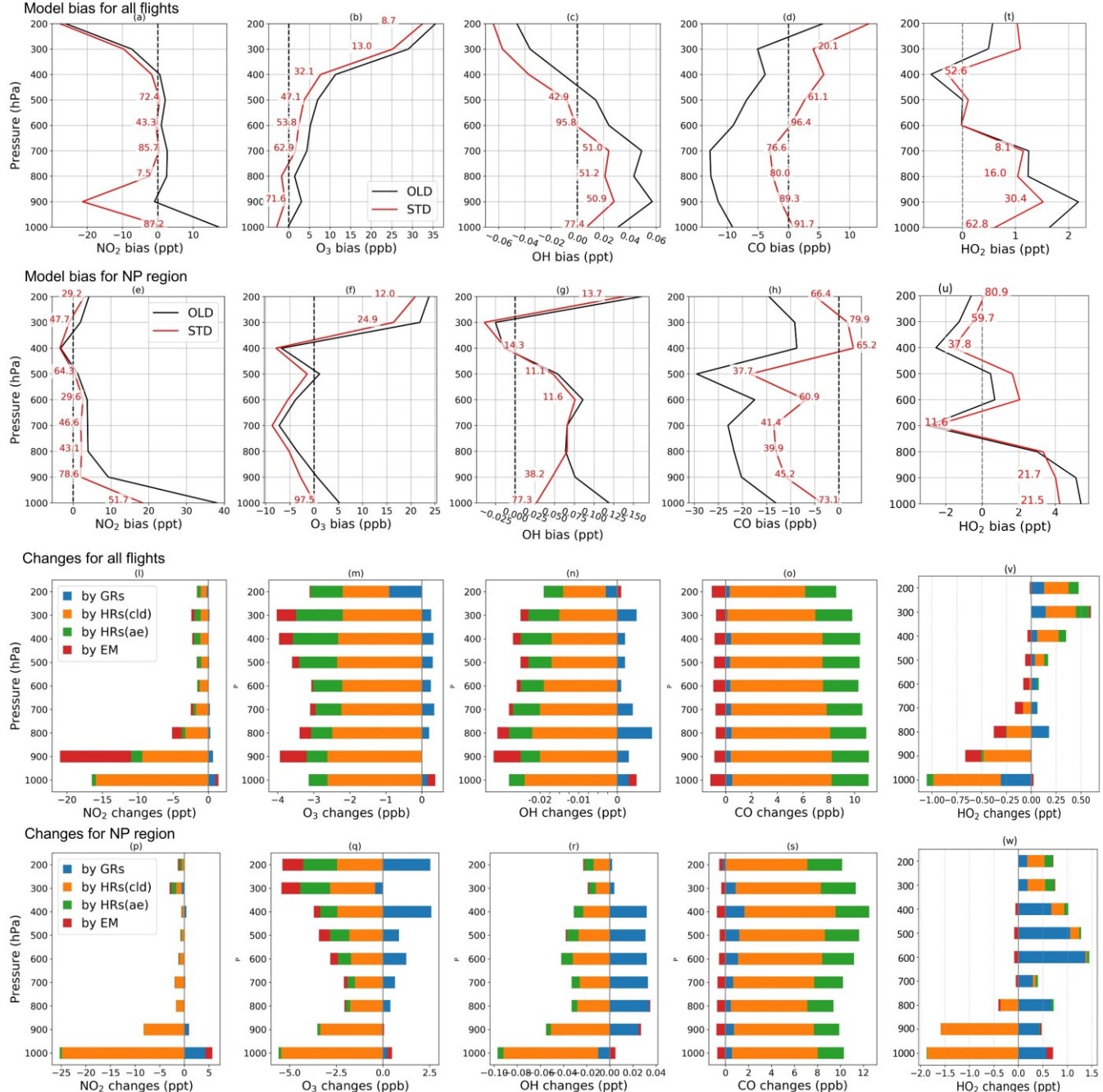

Figure 5: Vertical profile of model bias against aerial ATom1 data (a-h, t-u) and changes by HONO chemistry (l-s, v-w) for NO₂, O₃, OH, CO, and HO₂ (from left to right columns). Biases in OLD (black lines) and STD (red lines) runs are calculated for all flights (a-d, t) and NP region (e-h, u). The red numerical texts are the relative reductions (%) of the bias in the STD run compared to that in the OLD run. Changes by GRs (blue), HRs on clouds (orange), HRs on aerosols (green), and EM (red) are calculated for all flights (i-o, v) and NP region (p-s, w).

### 3.1.4. Surface O$_3$ and CO in the marine environment

The simulations was also compared with the *research vessel (R/V) Mirai*'s observation in the western Pacific Ocean for O$_3$ and CO. The interpolation of model results for six cruises across Japan–Alaska region (40° N– 75° N, 140° E–150° W) in 7-8-9/2015–2017 (summer), Indonesia–Australia region (5°–25° S, 105–115° E), and Indonesia–Japan region (10–35° N, 129°–140° E) were provided for the period of 12/2015 - 1/2016. All the measured and simulated data were provided, whereas the data for the NP region (40°–60° N) were analyzed separately, as discussed in Section 3.2. More detailed information about the *R/V Mirai* can be found in Kanaya et al. (2019). Furthermore, the model evaluation with *Mirai* for the OLD run can be found in Ha et al. (2021).

Table 6 shows correlation coefficients, which indicated that the STD simulation for CO and O$_3$ agreed well with *Mirai* ($R$ = ~0.6). However, these correlation coefficients were slightly worsened compared with the OLD case. Although the HONO inclusion mostly reduced the model bias for CO, especially in the NP region (-16.158 to -4.948 ppb), the model bias for O$_3$ was increased. The model biases exhibited negative trend for both CO and O$_3$ in the OLD case. This simulation pattern for O$_3$ in the NP region was in line with the OMI comparison (Sect. 3.1.1). This finding seemingly indicated an insufficient downward mixing process of O$_3$ in the free-troposphere or inconsistent surface deposition (Ha et al., 2021; Kanaya et al., 2019). However, the CO underestimations in the NP region might mark the inadequate CO emission in the HTAP inventory in CHASER (Ha et al., 2021). In Figure 6(a,c), overestimations of CO and O$_3$ were visible along Japan-Indonesia-Australia (Track-2) during the low episodes in December/January. Here, the larger model biases might account for the model's insufficient halogen chemistry (Kanaya et al., 2019; Ha et al., 2021). Figure 6 shows the model's percentage discrepancies for O$_3$ from *Mirai*'s data, except those from HONO concentrations interpolated for these regions. The underestimated simulations of O$_3$ were enlarged, especially in the Japan – Alaska region, being driven by the reduction effects in the STD case. In another way, these effects weakened the O$_3$ overestimates across the land areas, namely, over the region near Japan and Indonesia-Australia. Moreover, the higher HONO levels were identified for these off-shore data with up to 1.4 ppb abundances (Figure 6(c): red numbers). This high HONO level might underestimate an accurate level as a stronger reduction for O$_3$ was still required for the STD run (Figure 6(a): red marks).

The effects of the HONO chemistry along *the R/V Mirai* tracks exhibited various trends for each mechanism. Figure 7(b,d) illustrates O$_3$ and CO changes triggered by the HONO gas reactions (GRs), uptakes (HRs), and emission (EM). The gaseous reactions (blue bars) had mostly increased CO levels due to the reduced OH and O$_3$ levels (Figure S14). The gaseous mechanisms caused some reductions at the peak CO level because higher OH level from HONO photolysis near land domain or extra OH flux from stratosphere near 60 °N latitude could dilute CO. Furthermore, O$_3$ level was slightly increased due to GRs north of 60° N, as GRs was a source for NO$_2$ thus enhanced O$_3$ formation at these high latitudes (Figure S14). O$_3$ level was often decreased in the NP region since minor NO$_2$ increased and stronger OH reduction were seen for this region (Figure S14). The change tendencies in O$_3$ near land areas were varied (T2, T3) because the vertical effects to NO$_x$ and OH were stronger during DJF for this region (Figure S14). HRs, largely consuming NO$_2$, reduced O$_3$ (as large as 8 ppb), and increased

CO (~10 ppb) levels (Figure 7(b,d): orange+green). HRs, (particularly HRs on cloud surfaces, shown by orange bars), exerted the strongest contribution to the calculated changes in $O_3$ and CO among the three HONO pathways. This predominant cloud

effect was also prominent in the previous comparisons, especially EMEP (Figure 10(ii,jj): blue), thereby, indicating substantial effects of clouds at the mid-latitudes where the cloud SAD is higher (Figure S1). HRs on aerosols (green bars) had minor contributions during all cruises, despite they caused a marked increase in the $O_3$ concentrations off-coast Japan (track #3). It should be noted that this is not enough to explain the simulation bias with regard to the measurements. The additional HONO from direct emission (red bars) mainly increased $O_3$ and reduced CO concentrations, especially near land (latitude < 50° N).

This finding resonated with the comparison for continental stations (Figure 10(cc,ff,ii,jj): oranges). The overall effects of the HONO chemistry along *Mirai* cruises tended to reduce $O_3$ and increase CO levels. For the NP region, CO level increased and OH level reduction also can ameliorated the model performance. The improved model performance is evidenced from the comparison of the simulation with ATom1 aircraft data as well (Figure 5(g,h)). Thus, the strengthened underestimation of $O_3$ concentration in the NP region was unlikely driven by HR on cloud particles (Sect 3.1.1 and Ha et al., 2021). It was rather

related to the inconsistencies in the surface deposition of ozone. These inconsistencies were supported by empirical evidence as the negative bias in this comparison turns neutral or positive for the aircraft measurements in the same region (Figure 5(f): at 1000 hPa).

Overall, the comparisons between the model and and ATom1 *Mirai* might indicate that the HRs on cloud surfaces were the main contributing factor to the marine boundary's photochemistry, whose effects emerged during the ATom1 flights in the

marine atmosphere. GRs and aerosol HRs had a stronger impact on atmospheric chemistry at higher altitudes than the near-surface layer. Also, their effects should be enhanced through the additional $NO_x$ and HONO sources to reconcile the model simulations with the observations.

**Table 6:** Model comparison with *Mirai* cruises: no outlier filter is applied. $N$ is the available data for each calculation. Correlation coefficient ($R$, no unit) and biases (ppbv) in STD run are shown as bold if better than those in OLD run.

|  | CO<br>$N=4030$ | CO (40–60° N)<br>$N=1374$ | $O_3$<br>$N=3893$ | $O_3$ (40–60° N)<br>$N=1418$ |
|---|---|---|---|---|
| $R$(STD) | 0.690 | 0.586 | 0.568 | 0.618 |
| $R$(OLD) | 0.696 | 0.601 | 0.628 | 0.642 |
| bias (STD) | **4.087** | **-4.948** | -8.823 | -7.823 |
| bias (OLD) | -8.136 | -16.158 | -3.625 | -1.472 |

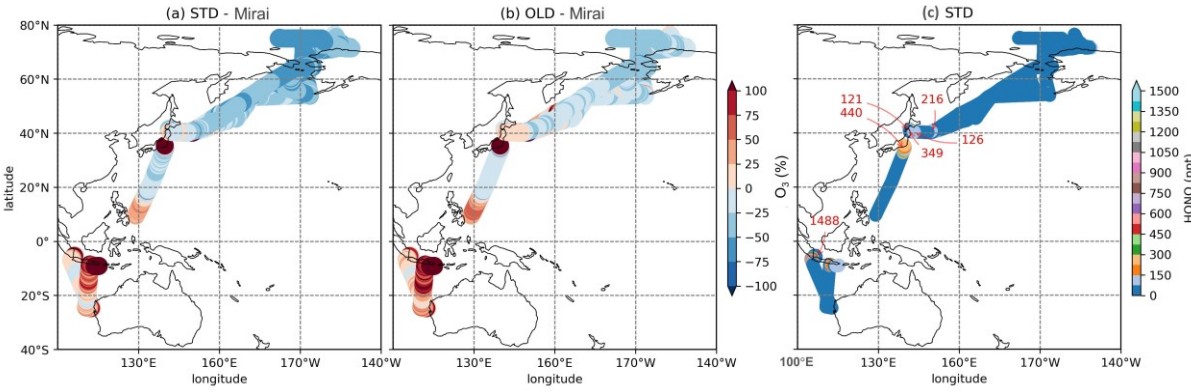


**Figure 6: Percentage discrepancies of STD (a) and OLD (b) simulations from *Mirai* for O₃ and HONO concentration in STD (c). The red numbers in (c) indicate maximum HONO concentrations for each cruise.**

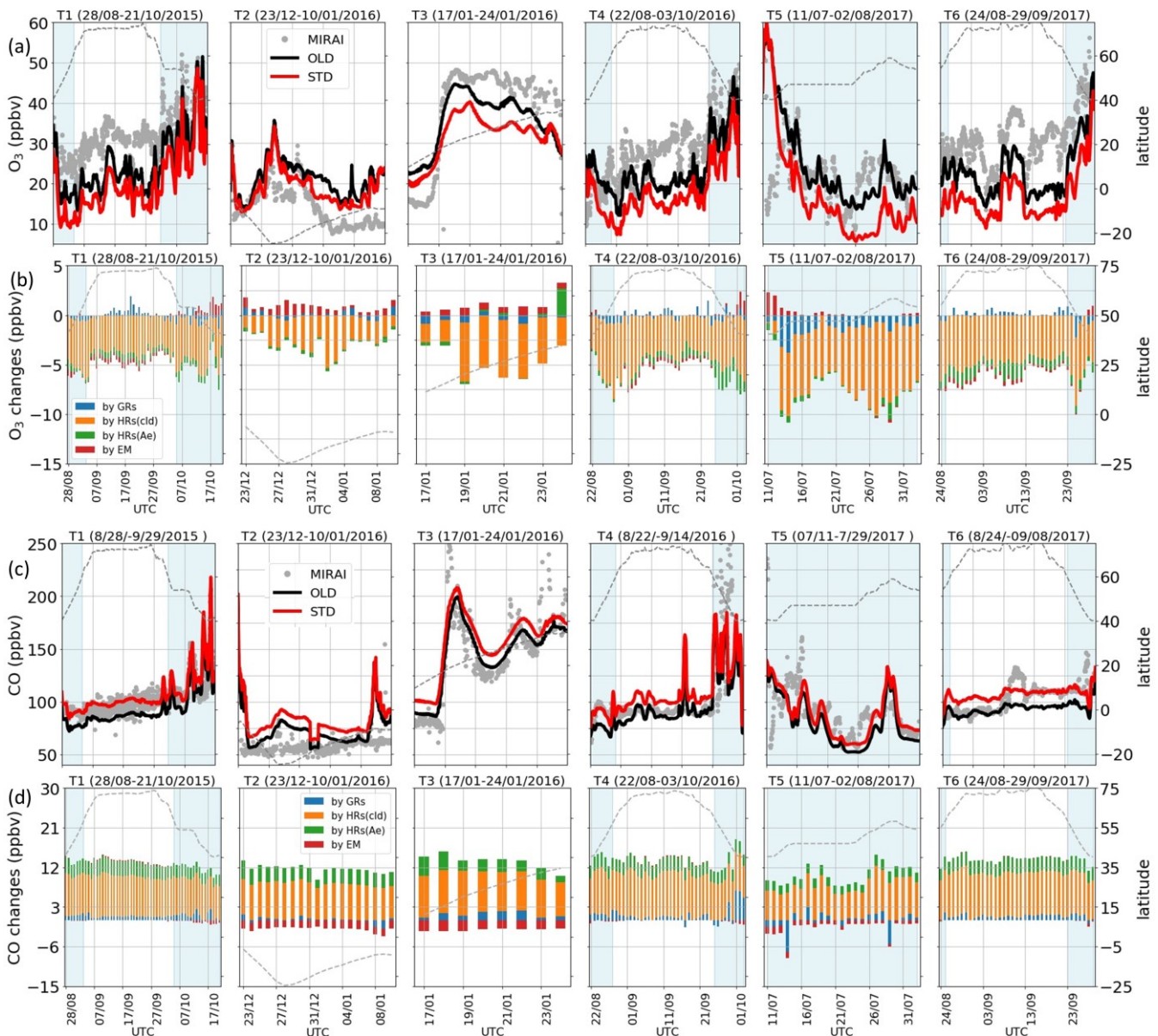

**Figure 7: Validation with ship-based data.** Observed and simulated concentrations (a, c) and daily mean effects by HONO chemistry (b, d) for $O_3$ and CO during *Mirai* cruises. In (a, c), grey dots: observation, black lines: OLD case, red lines: STD case. In (b, d), blue bars: changes by GRs; orange: by HRs on clouds; green: by HRs on aerosols; red: by EM. The left axis exhibits the concentrations and changes (ppbv). The right axis shows cruising latitudes plotted as dashed lines. The horizontal axis is travel times (UTC). Vertical light-blue shaded areas are for data in the NP region (140–240° E, 40–60° N).

### 3.1.5. HONO, NO$_x$, O$_3$, and other atmospheric species at ground-based stations

In this part, HONO concentration and HONO-related species measured for summer 2018 at the leg (150 m) and the summit (1,534 m) of Mt. Tai (Shandong province, China) by Xue et al. were reproduced using a data extraction tool from images (Figure 5 in Xue et al., 2022a). We compared our model's additional sensitivity simulations (Table 5) with Xue's measurements (Figure 8). The HONO estimated in the STD case for the foot station was rapidly reduced after 4 am (from ~ 1,100 pptv), while the observed HONO level peaks at ~ 6 am and remains about 0.5 ppbv at noon (Figure 8(a)). The HONO level produced

by the sole NO$_3^-$ photolysis on ground, aerosols and cloud particles (JANO3-A/B/C cases – respectively in solid blue, dashed cyan, and dashed orange lines in Figure 8(a)) can append moderately the simulated daytime HONO to reach the ground-based observatory levels comparing to the STD case. This append indicated a partially important role of NO$_3^-$ photolysis on all surfaces to HONO sources since early morning. Also, NO$_3^-$ photolysis adjusted the ratio of HONO/NO$_x$ more analogue to the observed daytime ratios at the ground (Figure 8(c,d)). The lack of NO$_2$ sources, especially during nighttime (Figure S15), even

in the OLD case (without HONO chemistry), was one of the reasons for the remaining unknown HONO sources existing during ~ 5-11 am. The combined cases (maxST+JANO3-A/B/C cases) produced too much HONO at the leg compared to observations (not shown in Figure 8), indicating the improper mechanism of enhanced NO$_2$ aerosol-uptakes for ground-based stations. In contrast, the enhanced aerosol uptakes were more compatible with observation at the summit, where these combined cases provided the best agreement to observation (Figure 8(a,b): solid cyan and orange lines). However, the best

simulation for HONO at the summit station only reached the lower line of the averaged-daytime HONO level (Figure 8(b)). Xue suggested that the high HONO level at the summit of Mt. Tai was dominated by the rapid upward transport from the ground and the in situ heterogeneous formation on the mountain surfaces (Xue et al., 2022a), which the mismatching between the actual locations and the coarse model grid (2.8°) of our model, including vertical layer, might not provide. Similar to NO$_2$, the simulated CO concentrations at Mt. Tai were very low (~200 ppb at the foot station in the STD case versus ~ 400-600 ppb

measured CO; not shown), even in the OLD case. In our model, the inadequate emission inventory of CO and NO$_2$ for the Asian region using HTAP-II-2008 and the coarse model resolution (2.8°×2.8°) were the reasons for the low ground-based emissions and vertical transport (Ha et al., 2021). Such emission inventory and vertical transportation improvement could further close the HONO observation gap and reduce the unrealistic highness of HONO/NO$_x$ ratios. Such a study could better show the validity of the HONO production mechanisms from enhanced NO$_2$ aerosol-uptake and NO$_3^-$ photolysis (combined

cases) for the summit station.

The above discussion confirmed that NO$_3^-$ photolysis (R7) on the ground, aerosols, and clouds surfaces (JANO3-A/B/C cases) enhanced daytime HONO but being ignorant of NO$_2$ level (Figure S15) due to an absent NO$_x$ recovery process (Ha et al., 2021). However, HONO sources via (R7) in JANO3-A/B/C cases still increased daytime O$_3$ levels at the foot of Mt. Tai (Figure 9(a)) as a result of rapid NO$_x$ cycling. Either the JANO3-A/B/C or the combined maxST+JANO3-B/C case was closest

to O$_3$ observation during cleaner (50-75 ppb O$_3$ on 16$^{th}$ – 22$^{nd}$ July) or uncleaner episodes (< 50 ppb O$_3$), suggesting an enhanced role of aerosol-uptakes during the polluted episode (indicated by the corresponding increases for CO; Figure 7 in Xue et al.,

2022a). At the summit station, NO$_3^-$ photolysis in JANO3-C and maxST+JANO3-C cases (all surfaces including clouds) boosted O$_3$ up to the observational level, indicating the contribution of cloud surface at the summit (~ 1,500 m).

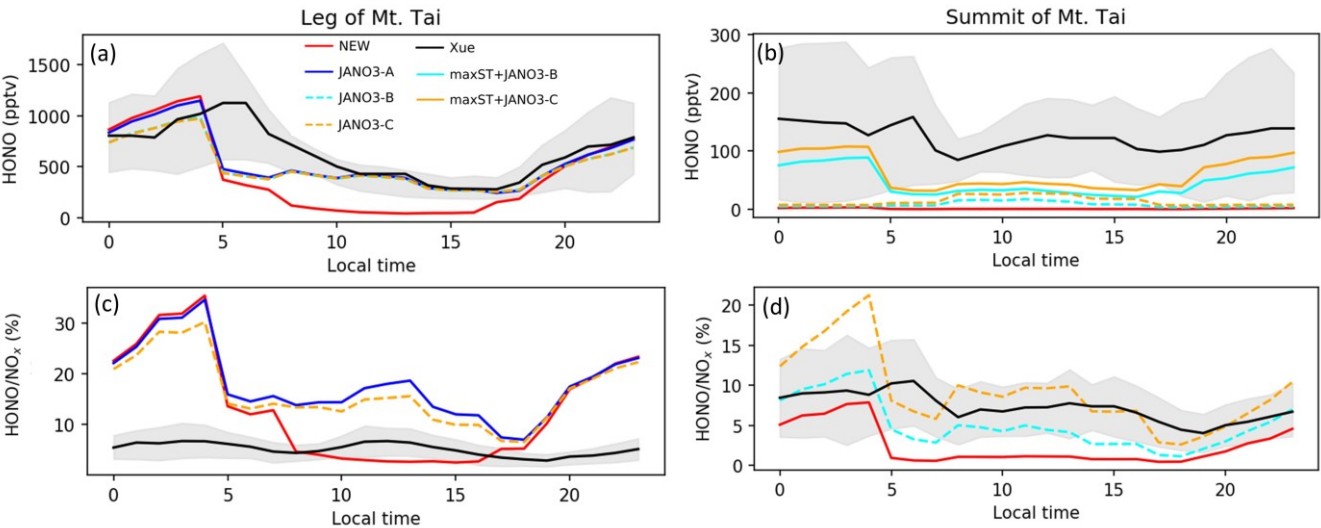

**Figure 8: HONO and HONO/NO$_x$ ratio concentrations estimated in the model and reproduced from Xue et al. (2022a) for Mt. Tai's foot and summit stations during July. Shades show minimum - maximum ranges, while colours show simulations as in legend.**

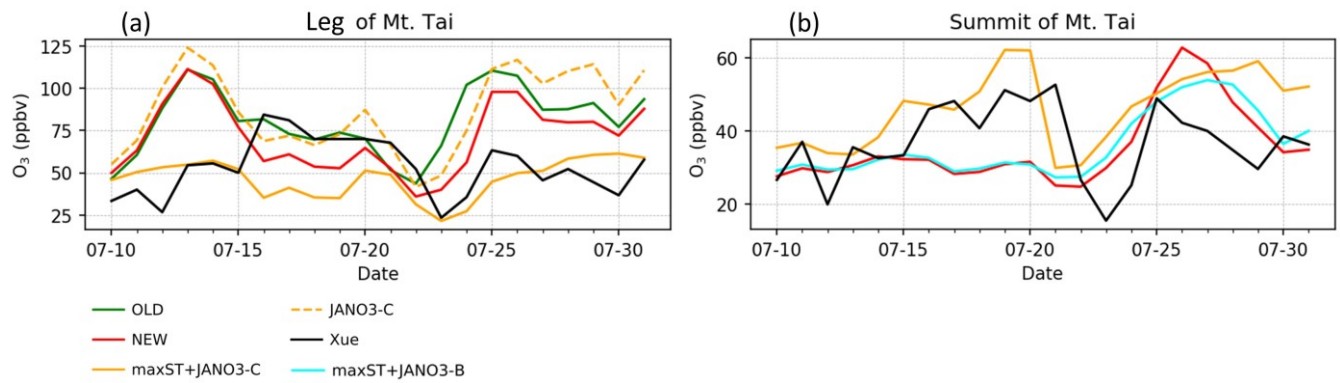

**Figure 9: Daytime (05:00–18:00) average O$_3$ concentration during 9-31 July 2018 at the foot and summit of Mt. Tai simulated in the model (coloured) and reproduced from Figure 7 in Xue et al. (2022a) (black). In (b), JANO3-B/C cases are overlapped with** 570 **maxST+JANO3-B/C cases, respectively.**

The effects of HONO chemistry in the continental near-surface layer of East Asia and Europe were also investigated. To this end, we conducted model comparisons versus EANET and EMEP stationary observations for mass and gaseous concentrations of PM$_{2.5}$, SO$_4^{2-}$, NO$_3^-$, HNO$_3$, NO$_x$, O$_3$, and CO (CO for EMEP only). Table 7 shows the correlation coefficients ($R$) and model biases for each species in the OLD and STD cases. The OLD simulation had its fair correlations and RMSEs 575 with observation for SO$_4^{2-}$ ($R$(EANET) = 0.56, $R$(EMEP) = 0.63), NO$_3^-$ ($R$(EANET) = 0.36, $R$(EMEP) = 0.71), and HNO$_3$ ($R$(EANET) = 0.18, $R$(EMEP) = 0.12), which were in line with other atmospheric chemistry models' $R$ and RMSE values against EANET and EMEP (Brian et al., 2017), as also discussed in Ha et al. (2021).

Figure 10 compares the measured versus simulated $HNO_3$, $NO_x$, HONO, $O_3$, and CO concentrations for the EANET and EMEP stations. The stations were divided into three groups: (1) high-$NO_x$ EANET stations, including Jinyushan (China), Kanghwa, Imsil, Jeju (South Korea), Bangkok, Nai Muaeng, Samutprakarn, Si Phum (Thailand), Metro Manila (Philippines), and Ulaanbaatar (Mongolia); (2) other EANET stations (39 for $HNO_3$, 22 for $O_3$, and 15 for $NO_x$); and (3) all EMEP stations. The ground-based observations in the period 2010–2016 revealed the slightly decreasing $NO_x$ for moderate $NO_x$ concentrations, as well as $PM_{2.5}$, and aerosols (Figure 10(e,h) and Figure S4(e,g,h,i)). These decreasing trends were not captured by our simulations which used the high emission scenario for the EDGAR/HTAP-2008 inventory. Note that $NO_x$ and $PM_{2.5}$ concentrations were generally underestimated in the model (OLD), especially in high-$NO_x$ regions (Figure 10(b,e,h) and Figure S4(a,d,g)) with model's averaged biases of -0.8 ppb $NO_x$ for EMEP and -4 ppb $NO_x$ for EANET (Table 7). These underestimations were stronger during winter, particularly for the high-$NO_x$ regions. It was possible that complex domestic sources could lead to diluted emissions for the simulations' moderate horizontal resolution (~2.8°). Higher model resolutions, such as 1.1°, 0.56° or even higher, could remedy such effects (Sekiya et al., 2018).

HONO chemistry in the STD case increased $HNO_3$, $NO_3^-$, $SO_4^{2-}$, and $PM_{2.5}$ for EANET and EMEP stations compared to the OLD case (Figure 10 and Figure S4: red vs blue lines). $HNO_3$ and $NO_3^-$ were increased as the products of $NO_2$ conversion (R4); thus, the model underprediction for $NO_3^-$ in EANET stations was mitigated (bias OLD→STD: -0.439→-0.223 μm m$^{-3}$). As a result of the increased OH level at the surface of these ground-based stations, $SO_4^{2-}$ was also increased (Li et al., 2015; Lu et al., 2018) (Figure S4(j,k,l)), although this effect enlarged the model overestimation for $SO_4^{2-}$ species at EANET and EMEP stations (Table 7). The consequent increase in $PM_{2.5}$, though minor, remedied the model underestimate for $PM_{2.5}$, e.g., model bias in OLD→STD: -3.044→-2.494 μm m$^{-3}$ (EMEP). Unfortunately, the model overestimate for $HNO_3$ in the OLD case was enlarged with the inclusion of HONO.

In the STD case, including HONO photochemistry, the negative biases of $NO_x$ in the model had adversely enhanced due to the $NO_2$ loss processes (bias OLD→STD: -3.997→-4.358 ppb for EANET (Table 7)). These processes also suppressed the $NO_x$ seasonality observed at most sites (Figure 10(b,h), red lines). The lack of seasonality was driven by the substantial loss of $NO_2$ on the surfaces of atmospheric particles during winter. For EANET's low $NO_x$ and EMEP stations, this huge $NO_2$ loss was attributed to cloud surfaces (Figure 10(ee,hh): blue bars). However, $NO_2$ uptake by aerosols has a comparable contribution effect to the cloud effect in high-$NO_x$ environments such as Jinyunshan (Figure 10(bb), grey bars). Namely, nearly half of the $NO_2$ was converted to $HNO_3$ in R4 (Figure 10(aa,dd,gg)) without an efficient recycling process, leading to an overall removal of $NO_x$. This lack of $NO_x$ could be the main driver for the seasonal $NO_x$ deterioration and the exacerbated overestimations of $HNO_3$ by simulations.

The STD $O_3$ simulation exhibited moderate and strong positive correlations with EANET and EMEP observations, 0.595 and 0.707, respectively (Table 7). The model improvements for $SO_4^{2-}$, $NO_3^-$, $PM_{2.5,}$ and $HNO_3$ were minor. However, the model improvement for $O_3$ was considerable, with a bias reduction of ~67% for EMEP and ~74% for EANET (Table 4). In the STD case, too little $NO_x$ was left from its heterogeneous loss, causing a net $O_3$ chemical destruction (because lacking atomic oxygen from $NO_2$ photolysis), which in turn, reduced the model overestimates for $O_3$ in the OLD case (Table 7; Figure 10(c,f,i), red

versus blue lines). However, further improvements in the chemical scheme were necessary to reproduce the $O_3$ measurements better; namely, a larger $O_3$ reduction for the summer and a reduced effect in simulated $O_3$ for the winter might alleviate the undesired effects. A delayed minimum from summer (as observed) to early winter (calculated in OLD and STD runs) causing
opposite seasonality for $O_3$ was prominent for the low-$NO_x$ EANET stations (Figure 10(f)). The effects of HONO chemistry on the mean OH levels were small, although it showed slight increases for OH's minima (Figure S4(j,k,l)). Thus, due to the apparent $O_3$ reduction for EMEP stations, CO was increased. Despite the reductions in $NO_x$ and $O_3$ levels being exaggerated during winter, the increment in CO reconciled the model's underestimation of CO high peaks in spring (Figure 10(j)), thereby strongly dwindling the bias for CO by ~59% (Table 4). However, the CO concentrations during summer should be reduced in
the STD case to capture the measurement. This finding might indicate inadequate HONO emissions for the EMEP stations (Figure 10(jj): oranges), which otherwise had reducing effects to $NO_x$, $O_3$, and CO levels during summertime.

The break-down scrutinies for aerosols and clouds effects for the ground-based stations (EANET/EMEP) also revealed the vast role of cloud-uptakes in the HONO impacts on $NO_3^-$ aerosols, $NO_x$, $O_3$, and CO (Figure 10 and Figure S16: blue bars), while the HONO impacts on $HNO_3$, $PM_{2.5}$, and $SO_4^{2-}$ aerosols were governed by aerosol-uptakes and HONO emission (grey
and yellow bars).

The existing ill-reproduction in $NO_x$'s seasonality and overestimations for $HNO_3$ might be amended by an explicit inventory for direct $NO_x$ emissions and an efficient $NO_x$-recycle process. Such mechanism via $HNO_3$ uptakes on soot surfaces ($HNO_3 \rightarrow NO_2$) was also tested in this study using the uptake coefficients range from $3 \times 10^{-5}$ - $4.6 \times 10^{-3}$ (Lary et al., 1997; Akimoto et al., 2019). Unfortunately, this heterogeneous $HNO_3$ conversion could not solely serve as a productive $NO_x$
recycling process in the EANET/EMEP stations (not shown). Among the additional cases described in Table 5, the alternated HONO : $HNO_3$ (0.9 : 0.1) product ratio of (R4) (ratR4 case) showed a good remedy for $NO_x$ at EMEP stations (Figure S11(g): brown vs black). For the EANET sites, the photolysis of adsorbed $HNO_3$ on ground surfaces (JANO3-A case) avoiding $NO_x$ removal via $HNO_3$ could remedy the $NO_x$ seasonality issue for these ground-based stations (Figure S11(a,d): greens vs blacks). However, the photolysis of adsorbed $HNO_3$ on grounds and aerosol/cloud surfaces (JANO3-B and JANO3-C cases) was not
an effective $NO_x$ recycling for EANET/EMEP measurements, leaving only slight differences in surface $NO_x$ levels compared to STD case (Figure S11(a,d,g): cyans and oranges vs reds). However, the heterogeneous photolysis of $HNO_3$ increased $O_3$ and OH at the high $NO_x$ regions instead of $O_3$ reduction in the STD case (Figure S11(b,c): cyans and oranges vs blues), which brought reconciliation to the underestimates for $O_3$ peak in springs (Figure S11(e): dotted cyan vs dotted blacks), although the runs with $HNO_3$ photolysis still did not capture the $O_3$ minimum in summer. Only the ratR4 case could capture the $O_3$ minimum
in summer among the sensitivity cases, which indicated the need for stronger $NO_x$ recycle processes for these ground-based stations.

Table 7: Model comparison of different species with observations at the EMEP and EANET stations. Three-sigma-rule outlier detection is applied for each station before calculating correlation coefficients $R$. $NO_x$ data are filtered once more using the two-
sigma-rule. $R$ and bias of the STD run are shown as bold if improved compared to the OLD run. Units for model biases for $PM_{2.5}$, $SO_4^{2-}$, $NO_3^-$: $\mu m\ m^{-3}$; $HNO_3$, $NO_x$, $O_3$, CO: ppb.

| | EMEP | | | | | | | EANET | | | | | |
|---|---|---|---|---|---|---|---|---|---|---|---|---|---|
| | PM$_{2.5}$ | SO$_4^{2-}$ | NO$_3^-$ | HNO$_3$ | NO$_x$ | O$_3$ | CO | PM$_{2.5}$ | SO$_4^{2-}$ | NO$_3^-$ | HNO$_3$ | NO$_x$ | O$_3$ |
| R(STD) | **0.528** | **0.655** | **0.755** | 0.115 | 0.637 | **0.707** | 0.526 | 0.344 | **0.560** | 0.359 | **0.228** | 0.202 | 0.595 |
| R(OLD) | 0.469 | 0.631 | 0.713 | 0.115 | 0.698 | 0.650 | 0.535 | 0.357 | 0.559 | 0.364 | 0.181 | 0.235 | 0.613 |
| bias (STD) | **-2.494** | 0.889 | 0.632 | **0.073** | -1.680 | **-1.708** | 2.943 | **-7.231** | 1.161 | **-0.223** | 0.396 | -4.358 | **1.410** |
| bias (OLD) | -3.044 | 0.752 | 0.247 | 0.077 | -0.818 | 5.154 | -7.138 | -7.583 | 0.981 | -0.439 | 0.302 | -3.997 | 5.494 |

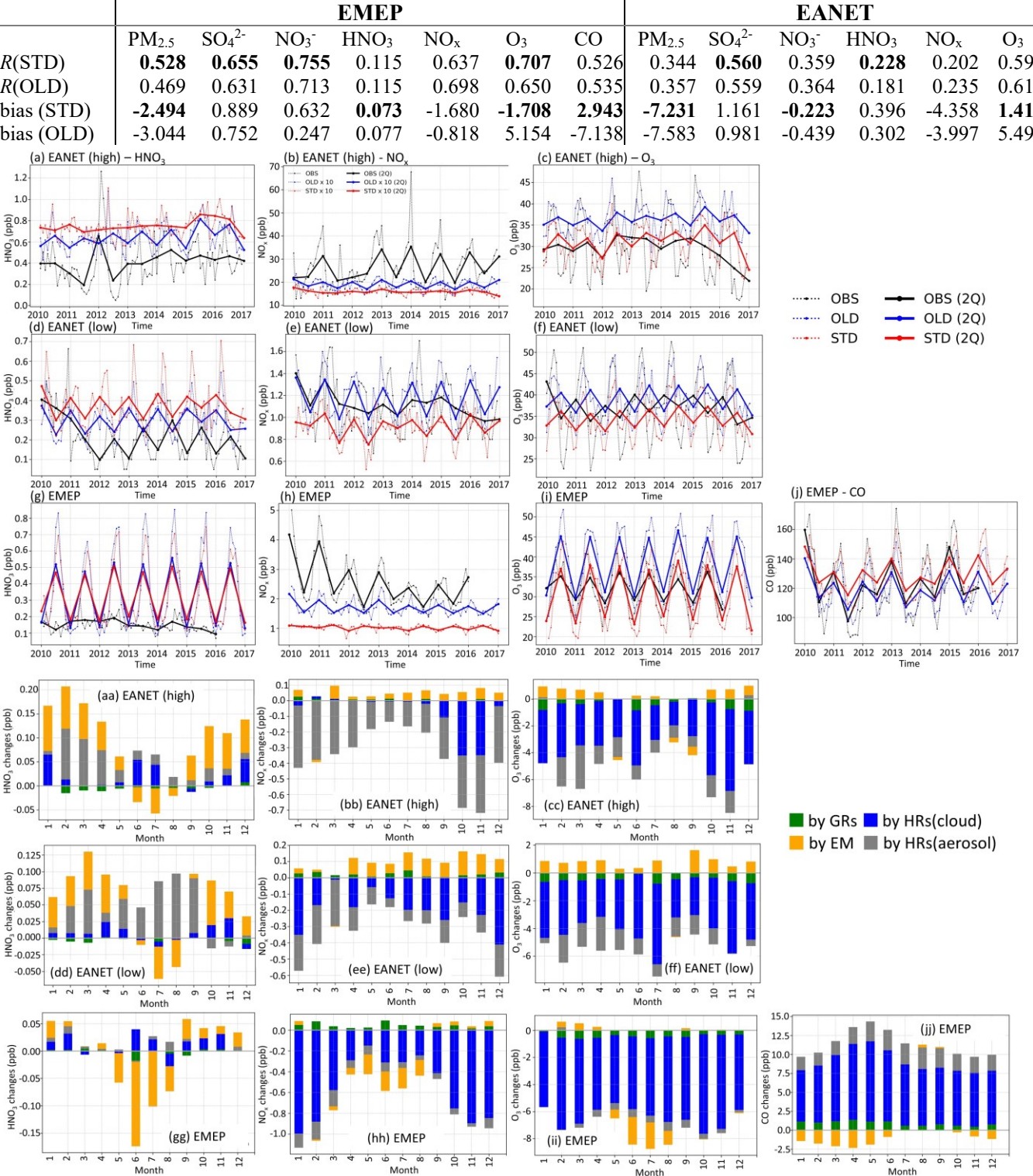

**Figure 10: Concentrations and changes by HONO inclusion for EANET and EMEP stations. (a-j) Observed and simulated concentrations during 2010–2016. Black lines: observation; red: STD case; blue: OLD case. In (b), concentrations in STD and OLD are 10 times-folded for better visualization (red and blue lines). For each group of stations, dotted lines are all stations' median from each station's monthly-mean values. Thick solid lines represent two quarters averaged from dotted lines. (aa-jj) Calculated monthly-mean changes by HONO chemistry. Green bars: monthly changes by GRs; blue: by HRs on clouds; grey: by HRs on aerosols; orange: by EM. Stations are grouped as high-$NO_x$ EANET (first and fourth rows), low-$NO_x$ EANET (second and fifth rows), and all EMEP stations (third and sixth rows). First column: $HNO_3$; second column: $NO_x$, third column: $O_3$, fourth column: CO.**

## 3.2. Distribution of HONO and global effects of HONO chemistry

### 3.2.1. Global HONO distribution and burden

This section sheds light on the global HONO distribution computed for the STD case. The surface HONO concentration was peaked over the geographical region that embraces China, with seasonal mean leveled up to 2.8 ppbv during summer and 7.8 ppbv during winter (Figure 11(a,b)). The winter peak agreed with observations for a large industrial region in the Yangtze River Delta of China (Zheng et al., 2020). The high concentrations of HONO were also identified in other industrial regional clusters: Northeast America (seasonal mean up to 0.5–1 ppbv), India (up to 1–3 ppbv), forest regions, especially the extra-tropic evergreen forest in Europe (up to 1–3 ppbv), and Africa (up to 0.5–1 ppbv). Over the ocean, HONO levels remained at 10–30 pptv in the coastal regions and below 10 pptv far off the coast. The simulated HONO distribution was in line with a previous study (Elshorbany et al., 2012) despite the peaks over polluted Chinese areas were markedly higher in our model (tenfold). The overestimation associated with the soot uptake in our model has been previously neglected. The highest HONO concentrations (10–30 pptv) in the free troposphere (at 2500 m) were simulated over Africa's biomass burning region during wet months (JJA) (Figure 11(c,d)), which could arise due to the $NO_2$ uptake on aerosols, originated from this wildfire source.

In the model, HRs and EM were the main contributors to HONO at the surface layer (Figure 12) by providing efficient HONO formation and promoting gas reaction (R2). Of the various surfaces provided for HRs in our model, liquid/ice cloud particle surfaces were supposed to catalyze significant photochemical effects in remote regions. This phenomenon has not been previously addressed in other studies in detail. The uptake of liquid/ice cloud particles either increased HONO formation via (R4) for the tropical and southern oceans or reduce it via (R6) along 60 °S in DJF and Artic in JJA (Figure 12(a,b)). Besides cloud particles, HRs on aqueous aerosols also produced HONO in a continental atmosphere rich in sulfate, dust, and soot particles (Figure 12(c,d)). EM included in the model has sharply increased the HONO level over deserts (Sahara, Arabian), grasslands (South Africa, South America), boreal, and agricultural land (West Europe, Australia). This finding agreed well with another study, based on spaceborne observations for HONO in wildfire plumes (Theys et al., 2020) and along ship tracks in the marine boundary layer (Figure 12(e,f)).

Table 8 summarizes the global sources and sinks of tropospheric HONO quantified by CHASER. The simulations indicated that GRs contribute only 11% of the HONO net production. HRs and EM produced more significant HONO (63% and 26% HONO net production, respectively). The pyrogenic HONO emission estimated in this study might be underestimated as HONO/$NO_x$ emission ratio could be enhanced by up to 1 at extratropic evergreen forests (universally 0.1 in this study

(STD)) (Theys et al., 2020). For large metropolitan areas such as those in China, HRs and EM had also been reported as the two most significant contributors to HONO formation, at ~59% and 26–29%, respectively (Li et al., 2011; Zhang et al., 2016). Of the various surfaces provided for HRs, aerosols represent a more effective HONO formation site (~51.2%) compared with ice and clouds, as they are contributing only 11.8% to HONO production. Moreover, the HONO loss through photolysis (R1) and (R3) was equivalent to its uptake onto the particles (R6). In equilibrium, the tropospheric abundance of HONO averaged of the globe was estimated to be 1.4 TgN in our model.

HONO production calculated in sensitivity cases (Sect. 3.1.2) are recorded in Table S5 (last column), and the spatial distributions are plotted in Figure S12. The small supplement by the photolysis of adsorbed $HNO_3$ on ground surfaces (JANO3-A case) to surface HONO concentration and tropospheric HONO burden (1.40 → 1.45 TgN) was consistent with the discussion for EMeRGe campaign. In the JANO3-B and JANO3-C cases, tropospheric HONO burden was increased to 2.02 TgN and 2.93 TgN, respectively, mainly remaining for the lower troposphere (~600 hPa) (Figure S12(g,h)). Compared to $HNO_3$ photolysis, the enhanced aerosol-uptake (maxST case) produced HONO more extensively over the source region and in the winter hemisphere where there was no photolysis (Figure S12(a-d)). Therefore, the maxST case did not produce enough HONO during EMeRGe (worse than JANO3 cases). However, the global HONO burden in maxST case was added to 7.79 TgN, which might be because we set the enhanced aerosol uptake of $NO_2$ for all environments. The combined cases (maxST+JANO3-B or maxST+JANO3-C), although appropriately approaching daytime HONO production as well as $NO_2$ and $O_3$ levels during EMeRGe, incredibly escalated the global HONO burden to 12.64 and 17.13 TgN, respectively (Table S5), more via the enhanced aerosol-uptake setting that could reach the upper troposphere (Figure S12(k,l)). However, it might be more realistic if HONO production stayed at the lower troposphere (Eshorbany et al., 2012). Further work for the combined cases to be standard cases, the amplified $NO_2$ conversion on aerosols should be confined in high SAD regions (Kalberer et al., 1999; Stadler et al., 2000), or at daytime only (Notholt et al., 1992; Stemmler, 2007).

**Table 8: Global sources and sinks of tropospheric HONO calculated by CHASER (2011).**

| Sources (TgN/yr) | 35.17 | Sinks (TgN/yr) | 33.77 | Net productions (TgN/yr) | 1.40 |
|---|---|---|---|---|---|
| $P_{GR}$ | 13.24 | $L_{GR}$ | 13.09 | GRs | 0.15(11%) |
| $P_{HR}$ | 15.31 | $L_{HR}$ | 14.43 | HRs | 0.88 (63%) |
| $P_{HR(cld)}$ | 9.57 | $L_{HR(cld)}$ | 9.40 | HRs(cld) | 0.17 (11.8%) |
| $P_{HR(ae)}$ | 5.74 | $L_{HR(ae)}$ | 5.03 | **HRs(ae)** | **0.72 (51.2%)** |
| $S_{EM}$ | 6.62 | $L_{EM}$ | 6.25 | **EM** | **0.37 (26%)** |

P denotes chemical production, S denotes source (emission + chemical production), and L denotes loss. The numbers in parentheses represent the portion of each pathway to the total HONO net production. Bold lines show the most significant contributing mechanisms to the HONO burden.

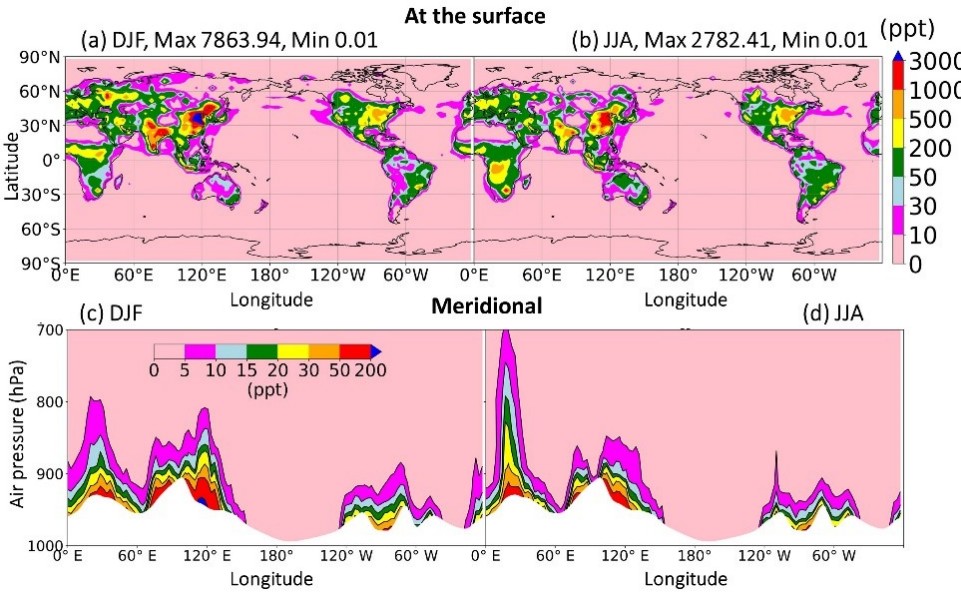

**Figure 11: Distribution of HONO levels at the surface (a-b) and meridional mean (c-d).**

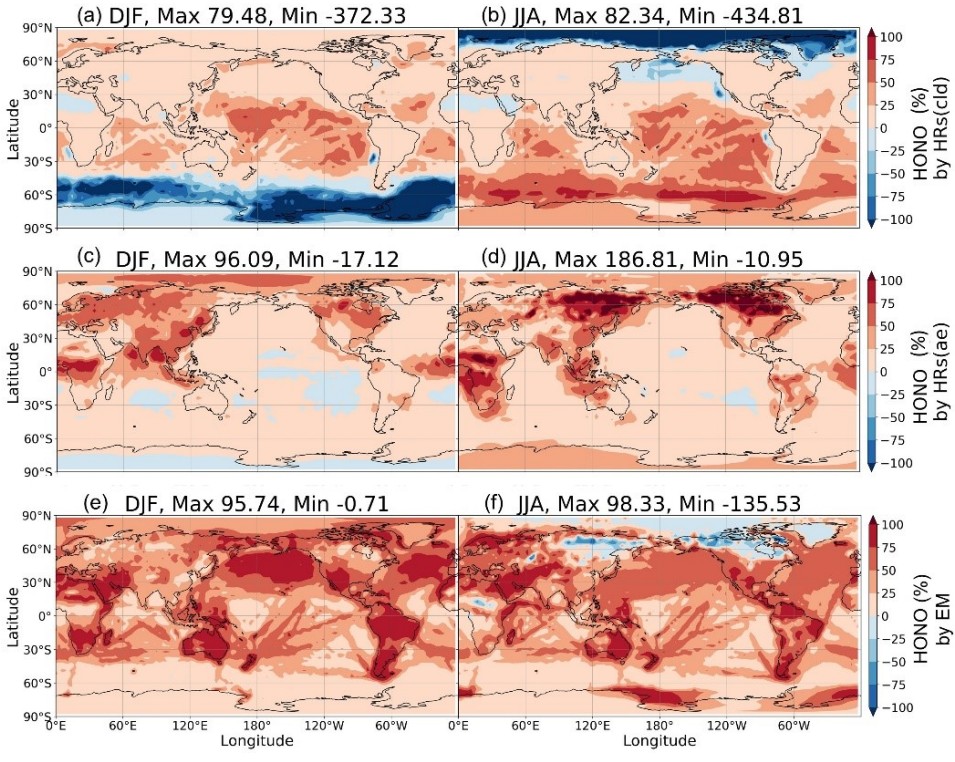

**Figure 12: Contribution of HRs and EM to surface HONO concentrations. Contributions of HRs onto ice and clouds (a, b), HRs onto aerosols (c, d), EM (e, f) in DJF (left) and JJA (right) are plotted. Each contribution is determined by the difference of HONO in two simulations: (a, b): GR+HR(cld) and GR, (c, d): GR+HR and GR+HR(cld), (e, f): STD and GR+HR, divide by HONO in STD case. The maximum and minimum values are out-scaled hence displayed at the tops of each panel.**

### 3.2.2. Global effects on tropospheric column ozone

Comparison between simulation and satellite OMI observation for tropospheric column ozone (TCO) can be examined as a global effect for ozone. In Figure 13, the STD run with HONO inclusion improved the overall tropospheric column ozone (TCO) distribution observed by the OMI, especially at the mid-latitudes. Figure S3 indicated a TCO reduction when HONO chemistry was included in the STD case (red lines vs green lines). Although HONO photolysis (R1) was a source of OH, supposedly increasing the tropospheric oxidizing capacity, the calculation in STD case showed the OH and $O_3$ increased only

occur at the surface of polluted sites (Figure 14(a,e)). The $NO_2$ conversion to HONO and $HNO_3$ (R4) became a $NO_x$'s removal pathway at remote regions, thus restricting the formation of $O_3$ and OH for the larger part of the troposphere (lacking atomic oxygen from $NO_2$ photolysis). Figure 13 shows that the $O_3$-reducing effects of HONO chemistry greatly reduced the model overestimates in the OLD simulation for the general Northern Hemisphere and polluted regions such as China. However, in the NP region, the inclusion of HONO only reduced the model overestimates during the insignificant episodes of TCO (autumn

to early winter) while extending the underestimates for TCO for the rest of the year (Figure S3(b)). These $O_3$ underestimates in the NP region are also visible for the modelled surface air versus the measurements during the *Mirai* cruises (Figure 6(a,b)). Notably, these underestimates for $O_3$ could hold up to 400 hPa, as seen in comparison with the ATom1 flights (Figure 5(f): 400–900 hPa). This phenomenon could be related to the stratospheric downward transport and insufficient vertical mixing, as discussed in Sections 3.1.3 and 3.1.4, for comparisons in the NP region's surface air and free troposphere. Although the HONO

level in STD remained < 10 ppt for this area (Figure 11), the $O_3$ reducing effects exacerbated the model discrepancy. The HONO photochemistry was unlikely to be the primary driver of this phenomenon as the ozone simulation was improved over the continents when the HONO photochemistry is included.

     In the combined sensitivity case maxST+JANO3-B (Sect. 3.1.2), $O_3$ was further reduced than the STD case. The reduction in TCO might be due to the enhanced $NO_2$ uptake on aerosols leading to more substantial $O_3$ formation restriction. The

maxST+JANO3-B case showed better harmony with OMI for the regions of TCO overestimations (Figure 13(g,h,k)), especially the annual mean (g panel). However, the underscores of TCO, including the NP region, were worsened. These results indicated that the reduction for $O_3$ by HONO chemistry was reasonable, and the combined cases such as maxST+JANO3-B could be plausible, although the estimated reduction degree should be reduced by elaborating the maxST's reactive conditions.

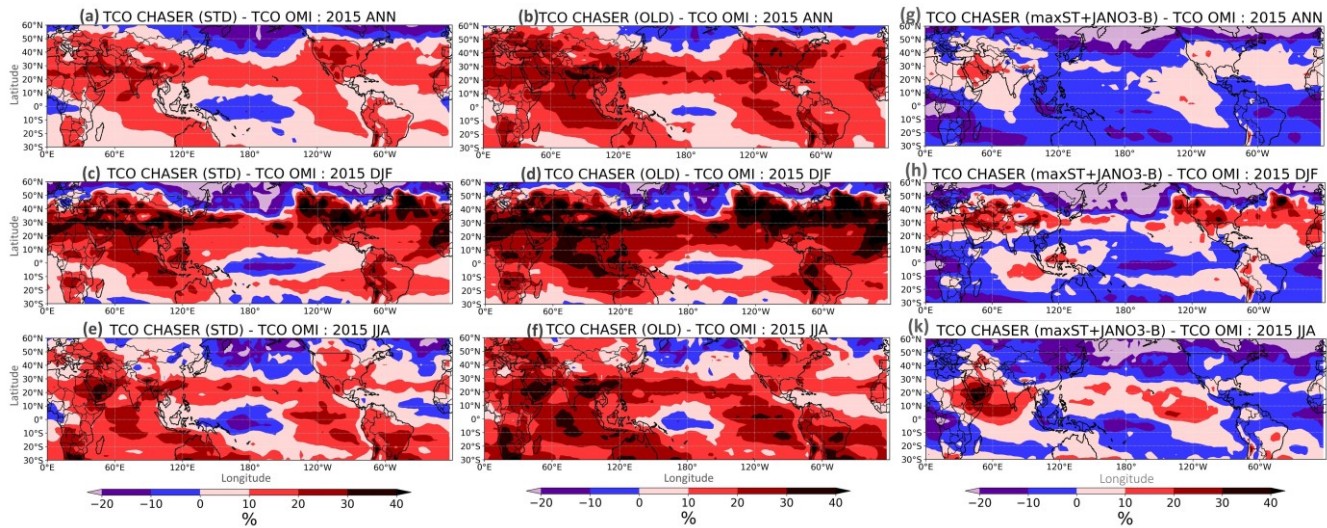


**Figure 13: TCO percentage differences between model and OMI. Left panels show STD versus OMI; middle panels show OLD versus OMI for annual (a,b), December – January – February (DJF) (c,d), June – July – August (JJA) (e,f). Right panels show differences for maxST+JANO3-B case versus OMI for annual, DJF, and JJA, respectively (g,h,k).**

### 3.2.3. Implication of HONO on the tropospheric photochemistry

In this section, the global impact of HONO photochemistry is elucidated. To this end, Table 8 summarizes the HONO budget and the contribution of each pathway to the HONO photochemical cycle. Table 9 describes its consequences for the lifetime of $CH_4$, and the budgets of $NO_x$, $O_3$, and CO. The gaseous reactions of HONO tended to increase the abundance of $NO_x$, $O_3$, and CO (+1.01%, +0.15%, +0.44%, respectively) and $CH_4$ lifetime (+0.36%) in the troposphere. Without heterogeneous and direct emissions, the relatively low HONO formation by gaseous reactions (11% of the total net HONO production; Table 8)

did not cause any significant effects on $NO_x$, $O_3$, and CO in the troposphere.

Heterogeneous reactions that produce HONO were the most salient contributing factors to tropospheric chemistry, thereby, decreasing the tropospheric oxidizing capacity and increasing the $CH_4$ lifetime by 15% and CO abundance by 10%. HRs also reduced $NO_x$ level by 23% and $O_3$ level by 6%, respectively (Table 9). The global HONO distribution from Figure 14 was mainly caused by the HR formation of HONO. Here, the reducing effects for $NO_x$ levels, with consequences for OH

and $O_3$ level reductions by heterogeneous reactions, were significant at mid-to-high latitudes during summer. More specifically, in DJF along 60 °S and the Arctic and NP oceans during JJA, which amounted to about a -100% reduction in $NO_x$ level at the surface (-60% in OH and -40% in $O_3$) (Figure 14(a-f): blue areas). These reductions in $NO_x$, OH, and $O_3$ levels extended up to 400 hPa at high N/S latitudes (Figure 14(k, l)). All these reduction effects for $NO_x$, OH, and $O_3$ levels were due to the removal of HONO on ice and cloud particles (R6) (Figure 12(a, b): blue fields). On the one hand, it accelerated the

conversion of $NO_2$ to HONO and ultimately strengthen its deposition by particulate nitrate (R4) (Figure S13(a)). On the other hand, HRs occurring on aerosol surfaces led to increments in OH and $O_3$ near the surface of polluted regions during winter.

These were the main contributors to the regional photochemical effects over China, Western Europe, and East U.S. regions in winter (up to -74% in $NO_x$, +1500% OH, +48% in $O_3$, Figure 14(a,c,e)). However, these OH and $O_3$ levels increase were only accumulated in the surface layer (only small red areas at ~1000 hPa in Figure 14(i,m)). Compared to HRs on aerosol, HRs on clouds exhibited twofold effectiveness when reducing tropospheric $NO_x$ level (-15% versus -8%) and caused threefold effects on the tropospheric oxidation capacity (+11.5% in $CH_4$ lifetime, -4.6% in $O_3$, +7% in CO), compared with HRs on aerosol (+3.5% $CH_4$ lifetime, -1.5% $O_3$, +2.6% CO) (Table 9).

Given the direct emissions of HONO (~10% of $NO_x$ emission inventory), the surface $NO_x$, $O_3$, and OH concentrations were generally enhanced in the STD case compared to the GR+HR case. They induced the concentration modification for $NO_x$ (+1.77%), $O_3$ (+0.97%), CO (-1.63%), and a significant reduction (-2.3%) in the $CH_4$ lifetime (Table 9). Remarkable enhancements for $NO_x$ (up to +198%), OH (+243%), and $O_3$ (+24%) (Figure 14(b,d,f): red fields) were identified for the cropland and shrubland/forest regions in Australia, South America, and South Africa during JJA, and the boreal vegetation prevailing at mid-high latitudes in Europe, North America, and the polluted Chinese region in DJF (up to +748% OH) (Figure 14(a): red fields). $NO_x$ and OH were elevated in these mid-latitude regions because of the enhanced HONO photolysis (R1) by the additional HONO source. However, OH, $NO_x$, and $O_3$ levels were reduced near the surface of the Northern Hemisphere's land during summer (up to -47% OH, -82% $NO_x$, and -15% $O_3$) (Figure 14(b,d,f)). The latter phenomenon was similar to the heterogeneous cloud effects for the high latitudes discussed above.

Overall, the inclusion of the three HONO processes (gas phase, aerosol and cloud uptakes, direct emission) caused -20% in $NO_x$, -5% in $O_3$, +8% in CO, and a significant increase of +13% $CH_4$ lifetime in the troposphere (Table 9). Figure 15 highlights the consequences of $HO_x$, $NO_x$, and $O_3$ for the Chinese and NP regions. $NO_x$ level reduction accumulated in the Arctic and Antarctic during summer, especially over the NP ocean (reducing $NO_x$ level by 60–90%, Figure 15(i)). These reductions in $NO_2$ and HONO concentrations were due to their uptake onto ice and clouds in these regions. However, these reducing effects caused further reductions in OH and $O_3$ levels for a larger part of the troposphere. As $NO_x$ was essential in regulating $O_3$ and OH in the troposphere, a reduction of $NO_x$ level increased the $HO_2$/OH ratio (due to the $HO_2 + NO \rightarrow OH + NO_2$ reaction), which restrained the formation of OH and ultimately of $O_3$. Moreover, a $NO_2$-deficit environment directly affected the $O_3$ level as $NO_2$ was a primary source of an oxygen atom that engages in the formation of $O_3$. Thus, in summer, both OH and $O_3$ levels were drastically reduced over the NP region (35–67% for OH, 30–43% for $O_3$, Figure 15(g,k)), and CO level was increased by 18% in this region (Figure 14(h)).

The significant impacts of HONO photochemistry were especially relevant over Eastern China in winter, which might reduce $NO_x$ level by 48–78% (Figure 12(c)) due to the uptake of $NO_2$ onto aqueous aerosols. At the surface, OH level was enormously increased as a result of HONO photolysis (R1), heterogeneous $NO_2$ conversions (R4, R5), and additional direct emissions (Figure 15(a,b)). The corresponding increase in $O_3$ level was only identified at the surface of the Beijing region during winter, with +28.8% caused by HRs on aerosols (Figure 15(e)). For Beijing with high $NO_x$ emissions, VOC-limited $O_3$ chemistry was likely the driving mechanism (Liu et al., 2010). The vast increases in OH and $O_3$ levels over Beijing in winter were basically in line with the present knowledge of HONO photochemistry (e.g., Lu et al., 2018). Elshorbany et al. (2012)

also reported an increase in OH ($2\text{–}5\times10^6$ molecules cm$^{-3}$) and O$_3$ (0.3–0.5 ppbv) concentrations over polluted regions in China during winter. Compared with Elshorbany's work, the increases in OH and O$_3$ concentrations in our model were higher due to the different HONO mechanisms applied in the two models, simply an averaged HONO/NO$_x$ ratio (0.02) in their model. In particular, our newly added heterogeneous reactions on cloud particles (R4) caused significant reductions in OH, NO$_x$, and O$_3$ levels in the NP region during summer, which their model did not cover. The overall reductions in tropospheric oxidizing capacity due to HONO photochemistry were in line with the expected response to heterogeneous processes (Liao et al., 2003; Martin et al., 2003) and agreed with those previously reported for other HRs (HO$_2$, N$_2$O$_5$, and RO$_2$) previously reported (Ha et al., 2021) (Table 9). Our findings indicate that a global model without heterogeneous processes for HONO would neglect the significant changes in OH and O$_3$ concentrations in remote areas and, thus, will underestimate the potential effects in polluted regions.

As mentioned above, the relative importance of ice and cloud surfaces to the oxidant chemistry was negative for Artic and NP regions. NO$_2$'s uptakes on ice and cloud surfaces (R4) were the main reason for the reductions in surface NO$_x$, OH, and O$_3$ concentrations in these regions during JJA (Figure S13(a)). These reductions also occurred for the free troposphere, which generally improved the model comparison with ATom (Figure 5) and partially improved in CO simulation by cloud effect (Figure 7). To HONO formation, enhancement of NO$_2$'s uptake coefficient on cloud surface ($\gamma_{liq.}$(R4) = $10^{-4}$ → $10^{-2}$), along with changing HONO/HNO$_3$ yield ratio from 0.5:0.5 to 0.9:0.1 in (R4) in the ratR4+CLD case compared to STD case helped more NO$_x$ preserved during EMeRGe's flights. Here, a supplement for HONO production is only seen in the ratR4+CLD case for the marine environment ~ 1,000 m near Japan (Figure 3(g): red-bordered orange diamonds). The ratR4+CLD case introducing an approach to recycle NO$_x$ led to lowered global effects of HONO (only -8.57% NO$_x$ globally, CH$_4$ lifetime = 9.6 years; Figure 16). For the ground-based station in comparison with Xue's data (Xue et al., 2022a), the NO$_3^-$ photolysis in JANO3-C and maxST+JANO3-C cases (all surfaces including clouds) boosted O$_3$ level, which improved the agreement with observed O$_3$ at the Mt. Tai's summit station, indicating the contribution of cloud surface to O$_3$ formation at the altitudes ~ 1,500 m over a mountainous area (Figure 9(b)). The NO$_3^-$ photolysis on cloud surface in the JANO3-C case could also be an effective HONO supplement for the tropospheric part above 2,000 m over the Asian coastal region compared with EMeRGe's data as compared to JANO3-B case, which excluded clouds (Figure 2(a)). The simulated concentrations of NO$_2$ and O$_3$ also agreed better with the measured data in this comparison.

Contrary to cloud surfaces, the aerosol effect was only crucial for regional photochemistry at the surface layer of polluted regions, such as China, Western Europe, East U.S. in winter time (Figure 14). As discussed above, aerosol uptakes reduce NO$_x$ but increase regional OH and O$_3$ levels. For the sensitivity of HONO formation to aerosol effect, the cases JANO3-B and maxST+JANO3-B, which only included ground and aerosol surfaces for NO$_3^-$ photolysis, also remedied the discrepancies for daytime HONO level across various altitudes during EMeRGe flights (Figure 2(a)). For the comparison with Mt. Tai station, enhanced uptakes of NO$_2$ onto aerosol surfaces in the combined cases (maxST+JANO3-B/C) provided more HONO production at the summit (Figure 8(b)), as well as adjusting O$_3$ levels at both foot and summit stations during the polluted episode (Figure 9).

The estimated global effects of HONO chemistry in the STD case was the abatement of global tropospheric oxidizing power, despite surface OH and $O_3$ levels being increased at polluted sites. The reduction tendency in global OH and $O_3$ contrasted with other modelling studies (e.g. Elshorbany et al., 2012; Jorba et al., 2012; Lee et al., 2016; Zhang et al., 2021). Some discussions on the tendency of HONO's global effects are addressed here. The positive or negative impacts on oxidizing species (OH and $O_3$) were constrained to HONO formation mechanisms rather than $NO_x$ concentration. For high $NO_x$ regions

such as EANET stations with six-months-averaged $NO_x$ higher than 20 ppb, $O_3$ and OH were reduced due to $NO_x$ removal via $NO_2$ uptakes (Figure 8(b,c)), especially at night. When $NO_x$ was highly underestimated in the model for these high $NO_x$ regions, and an efficient $NO_x$ recycle process was still absent, OH and $O_3$ might be reduced daily. The calculation for daytime-only in comparison with Xue'data showed that $O_3$ could be increased when a compliment HONO source was provided via $NO_3^-$ photolysis (Figure 9(a)) and sole enhancement of aerosol-effect rather than both aerosols and clouds (JANO3-C versus JANO3-

B, Figure 9(b)), even though $NO_2$ was still underestimated with a large extent in the model (Figure S15). At higher altitudes over remote regions in the ATom ($NO_2 < 10$ ppb) and EMeRGe comparisons, OH and $O_3$ can be increased due to more potent gas-phase chemistry of HONO in STD case, including HONO photolysis (Figure 4) and particle $NO_3^-$ photolysis (Figure 2).

       The amplified aerosol uptake of $NO_2$ (maxST case) further reduced an unrealistic degree of global $NO_x$ abundance (-55.4%) and tropospheric oxidizing capacity, leading to 14.5 years for a global $CH_4$ lifetime. The ratR4+CLD case introducing

an approach to recycle $NO_x$ led to lowered global effects of HONO (only 8.57% $NO_x$ was reduced globally, $CH_4$ lifetime = 9.6 years). The photolysis of adsorbed $HNO_3$ on ground surfaces (JANO3-A case) still showed reductions in global OH and $O_3$ abundances (Table S5). The ground-surface $HNO_3$ photolysis in the JANO3-A case caused only minor changes for a thin surface layer which is in line with other studies (Ye et al., 2018; Zhang et al., 2009). In JANO3-B and JANO3-C cases, a recycling process for $NO_x$ via $HNO_3$ photolysis was expected. However, only the JANO3-C case showed an increment in

global $NO_x$ and $O_3$ (+29% and +16.1%, respectively), leading to only 5.4 years for the global $CH_4$ lifetime, which was impractical. This was because that simplified approach and maximum thresholds for the phase $HNO_3$ photolysis were used. The combined case maxST+JANO3-B led to more convincing effects ($CH_4$ lifetime was 10.2 years; Table S5), which held the same tendencies as those calculated in the STD case. However, validating this combined case was only conducted for the daytime environment during EMeRGe (Sect. 3.1.2) and for TCO at northern mid-latitudes with OMI (Figure 13).

Figure 16 illustrates the calculated global-mean changes of tropospheric abundances in additional simulations from those in the OLD case (without HONO chemistry). The simulations of the largest negative to largest positive magnitudes of changes (%) in $CH_4$ lifetime (purple bars) were shown from left to right. Other bars show percentage changes in $NO_x$ (red), $O_3$ (blue), and CO (green). In simulations including ratR4, ratR4+CLD, JANO3-A, maxST+JANO3-B, and maxST cases, the HONO's impacts on tropospheric $CH_4$ lifetime and abundances of $NO_x$, $O_3$, CO showed similar tendencies to those impacts in the STD

simulation. These similarities indicated that the heterogeneous chemistry of HONO has a general tendency to reduce tropospheric oxidizing capacity (OH and $O_3$) as a result of $NO_x$ removal globally via $NO_2$'s uptakes on aerosols and clouds. In these settings, more substantial reductions in oxidizing species and $NO_x$ were seen in maxST and maxST+JANO3-B cases via enhanced aerosol uptakes of $NO_2$. The particle-phase $NO_3^-$ photolysis can solely compensate $NO_x$ removal processes and act

as an efficient $NO_x$ recycling mechanism (on a global scale) which can be seen in the JANO3-B and JANO3-C cases. The enhanced aerosol uptake in the maxST setting of the combined maxST+JANO3-C case neutralized this compensation. However, the tropospheric oxidizing capacity (OH and $O_3$) was increased in the cases configured with $NO_3^-$ photolysis (JANO3-B/C, maxST+JANO3-C cases), leading to a reduction in global $CH_4$ lifetime. The impacts quantitive was unrealistic in some cases, e.g., 30% $CH_4$ lifetime in maxST+JANO3-C case or -50% global $NO_x$ in maxST+JANO3-B case, proven as the proper mechanism for a particular environment (along EmeRGe-Asia-2018 flights) not globally. Thus, these changes merely provided the tendencies of impact sensitivity for different pathways of HONO formation, still with high uncertainty in their magnitudes.

In conclusion, we suggest the global effects tendency was towards tropospheric oxidizing capacity reduction, although further elaboration for enhanced aerosol-uptakes of $NO_2$ and surface-catalyzed photolysis of $HNO_3$ could drive the effect magnitude. The implication of HONO chemistry in a bottom-up approached global model such as CHASER securely needs an intense examination of possible HONO sources and profound evaluations with observed HONO in the troposphere.

**Table 9: $CH_4$ lifetime and tropospheric abundances for $NO_x$, $O_3$, CO and the changes by HONO chemistry**

| Simulation ID | $CH_4$ lifetime | Abundances of tropospheric | | |
|---|---|---|---|---|
| | (yr) | $NO_x$ | $O_3$ | CO |
| | | (TgN) | $(TgO_3)$ | (TgCO) |
| OLD | 9.09 | 0.119 | 408.79 | 327.20 |
| GR | 9.12 | 0.120 | 409.38 | 328.65 |
| GR+HR | 10.49 | 0.092 | 384.25 | 359.90 |
| GR+HR(cld) | 10.17 | 0.102 | 390.46 | 351.53 |
| STD | 10.28 | 0.094 | 388.21 | 354.57 |
| Effects | Changes (%) | | | |
| by GRs: | +0.36 | +1.01 | +0.15 | +0.44 |
| by HRs: | +14.99 | -23.19 | -6.15 | +9.55 |
| by HR(cld): | +11.52 | -15.28 | -4.63 | +6.99 |
| by HR(ae): | +3.47 | -7.91 | -1.52 | +2.56 |
| by EM: | -2.30 | +1.77 | +0.97 | -1.63 |
| Total: | +13.05 | -20.40 | -5.03 | +8.36 |
| by HRs($N_2O_5$,$HO_2$,$RO_2$) (Ha et al., 2021) | +5.7 | -3.87 | -2.91 | +3.43 |

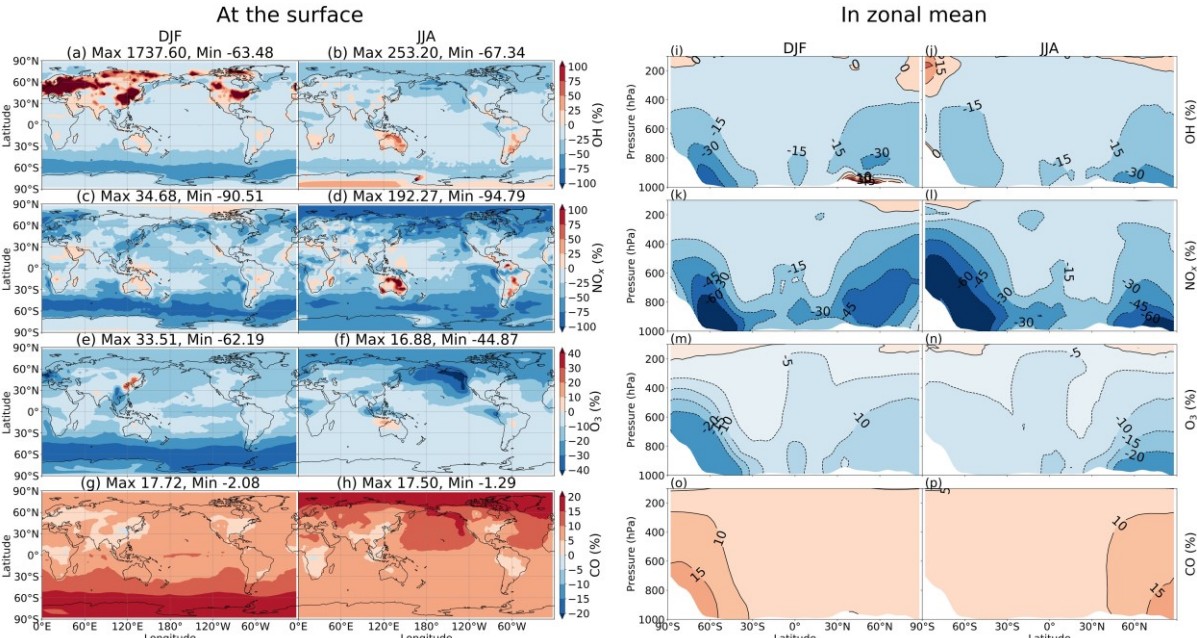

**Figure 14: Effects of the HONO photochemistry on the tropospheric oxidants OH (first row of panels), NOₓ (second row of panels), O₃ (third row of panels) and (CO last row of panels). Effects at the surface (a-h) and zonal means (i-p) are shown.**

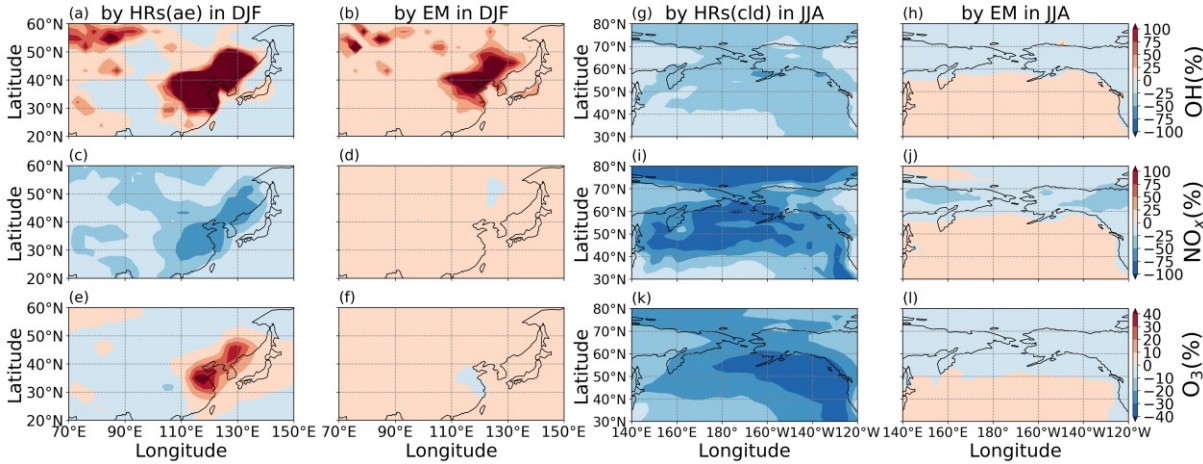

**Figure 15: Effects of HONO photochemistry for the surface layer, for OH (upper panels), NOₓ (middle panels), and O₃ (lower panels) over northeastern China region in DJF (a-f) and NP region in JJA (g-l) from dominant pathways of HONO by heterogeneous reactions of aerosols (1ˢᵗ column heterogeneous reactions ice and clouds (3ʳᵈ column) and direct HONO emission (2ⁿᵈ and 4ᵗʰ columns).**

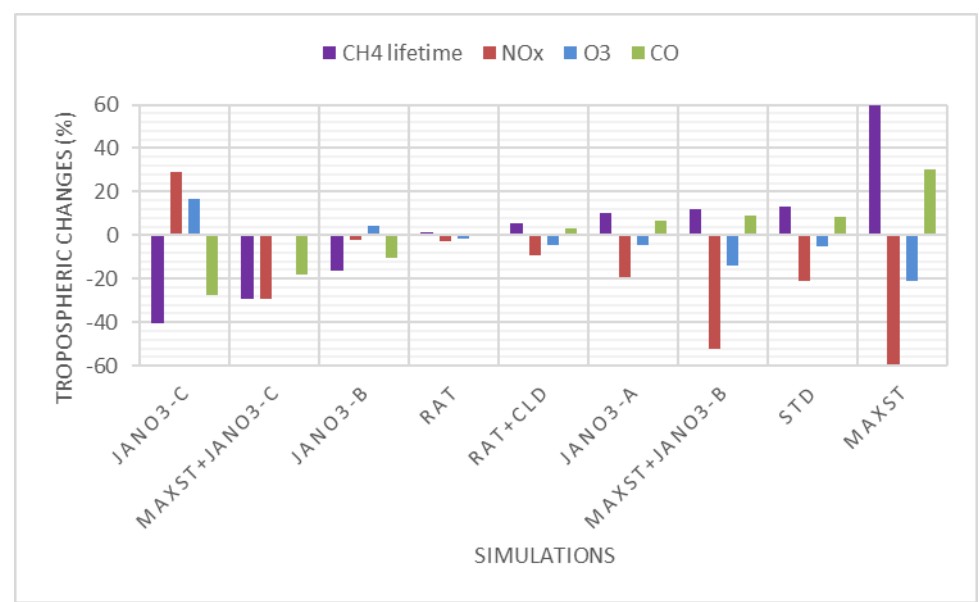

**Figure 16: Calculated global-mean changes of tropospheric abundances in additional simulations compared to OLD case (without HONO chemistry). From left to right, the order of shown simulations follows the percentage change magnitudes in CH₄ lifetime (largest negative change to largest positive change; purple bars). Other bars show percentage changes in NOₓ (red), O₃ (blue), and CO (grey).**

## 4. Conclusion

The HONO photochemical processes, including (a) the gas-phase reaction involving HONO, (b) direct HONO emission from combustion and soil crust, as well as (c) heterogeneous processes involving HONO were added to the chemistry-climate model (CHASER), which did not consider HONO chemistry before. We compared the measurements during the EMeRGe flights off the coastal region of East Asia and discerned good agreement between the measured and simulated $NO_2$, $O_3$, and CO profiles. However, the model does not reflect the influence of the Chinese river delta regions, as the large reductions in air masses affected by land emissions were identified. The model also stood out with $NO_2$, OH, $HO_2$, and CO improvements in the NP region, compared with the observations made during *Mirai* and ATom1, although the simulation underestimations of surface $O_3$ in this region was associated with the inconsistent surface deposition or vertical fluxes (c.f. from the stratosphere) becoming strong. We found that the model biases were reduced against the EANET/EMEP stationary observations for $PM_{2.5}$, $NO_3$ components, $O_3$, and CO concentrations when the HONO photochemistry was included.

In the model, the tropospheric abundance for HONO was 1.4 TgN and was made by 26% from the direct emission and 63% by HRs, in which HRs on clouds caused 11.8% and HRs on aerosols caused 51.2%. The HONO concentrations over the continents ranged from 30 ppt to 7 ppb and were maximized due to HRs over eastern China during winter. Only 5–10 ppt of HONO could be transported up to ~2000 m, indicating that its impacts remained mainly in the planetary boundary layer. We

argue that these simulated HONOs might underestimate the actual concentrations off-coast of eastern Asia in spring 2018. The unknown daytime HONO concentrations of up to 200 ppt measured in the boundary layer and free troposphere during the EMeRGe campaign were not reproduced by the STD simulation. Fortunately, the measured HONO was moderately captured by the combined simulation, which enhanced aerosol-uptakes of $NO_2$ and heterogeneous photolysis of $HNO_3$ (maxST+JANO3-B case). However, the enhancement for $NO_2$ uptakes on aerosols should be confined to particular environments to eliminate the effect exaggeration. Moreover, a further improvement of the model performance for the HONO photochemistry requires (1) the revised model's emission inventory with the emission sources of $NO_x$ and CO from South-Eastern and Eastern Asia, (2) the lighting-related $NO_x$ module is upgraded, and (3) the vertical mixing and downward fluxes from the stratosphere to be elaborated.

One or more renoxification mechanisms converting $HNO_3$ into $NO_x$ should be added to the model to overcome the observed and simulated $NO_x$ seasonality mismatches. Shifting the product ratio towards more HONO and less $HNO_3$ in reaction R4 could also provide more HONO and mitigated the deteriorated representation of $NO_x$ seasonality. The sensitivity tests also suggested that more robust aerosol processing in polluted areas and less $HNO_3$ product in R4 could further reduce $O_3$ level in summer, reducing the bias against measurements. The photolysis of adsorbed $HNO_3$ on ground surfaces (JANO3-A case) could also serve as a recycling process for $NO_x$ at Asian ground-based sites (EANET).

As calculated in the STD case, HONO chemistry reduced the global tropospheric oxidizing capacity, including OH and $O_3$ levels in global scale. It should be underlined that this finding is rather unexpected and contrasts with the increasing oxidation capacity previously reported for polluted areas. However, the global reduction effect to $O_3$ reduced the overestimations of OMI-based TCO by simulations, which notably included the geographical region embracing China. Of the three HONO sources, HRs produced the most prominent effects on the tropospheric photochemistry: reducing OH, $NO_x$, $O_3$, and increasing CO levels in the troposphere, leading to a +13.05% longer $CH_4$ lifetime, and -20.4% less $NO_x$, -5.03% less $O_3$, and an increased CO (+8.36%) abundance. In winter near the surface, gas-phase reactions involving HONO and $NO_2$ conversions on soot induced significant photochemical effects over eastern Chinese with changes of -60% in $NO_x$, +1700% in OH, +33% in $O_3$. During summer, HRs on ice and cloud particles could cause significant changes of -67% in OH, -45% in $O_3$, -75% in $NO_x$, and +17% in CO in the NP region. Albeit the more significant contribution of aerosols' heterogeneous reactions to the net HONO production, the heterogeneous processes involving ice and cloud particles were more significant globally. Our results from sensitivity tests demonstrated that the tendencies and magnitudes of HONO's global effects debated along with the effort regarding daytime HONO formation mechanisms. In capturing HONO measurement during EMeRGe campaign, the combined case enhancing $NO_2$ aerosol uptake and implementing heterogeneous photolysis of $HNO_3$ (maxST+JANO3-B) still resulted in the reduction for global tropospheric oxidizing capacity. In this case, the effect magnitude was smaller for $CH_4$ lifetime, but those for the $NO_x$-$O_3$-CO chemistry were stronger compared with the calculation in the STD case. Overall, our results proved that a global model without heterogeneous HONO formation, especially photochemical heterogeneous HONO formations, could bias the overall impacts of HONO on tropospheric photochemistry as it neglected the photochemical effects of HONO in remote areas and underestimated them in polluted regions. Our new finding on the

tropospheric oxidizing capacity reduction may affect climate change mechanisms and, as a result, may influence its mitigation policies.

## Code availability

The CHASER V4.0' source code and input data to recreate this work's results can be acquired from the repository at
https://doi.org/10.5281/zenodo.4153452 (Ha et al., 2020).

## Data availability

The primary data from *R/V Mirai* cruises for the period 2015–2017 are available from http://www.godac.jamstec. go.jp/darwin/e (last access: 30 September 2021). Due to a recent data security incident, the data owner (JAMSTEC) is suspending public access to this dataset. For any inquiries, please send an email to yugo@jamstec.go.jp. The data collected by
HALO aircraft during the EMeRGe campaign are listed on https://www.iup.uni-bremen.de/emerge/home/halo_payload.html and can be acquired via email to Lola Andrés Hernández <lola@iup.physik.uni-bremen.de>.

## Author contribution

HP composed all simulations and text. SK has the model code and supervised the findings of this study. KY and TF provided *R/V Mirai* ship data. HM, BS, and KP provided EMeRGe-Asia data. All authors have equally contributed to the discussion
provided within the manuscript and post-writing formatting and revisions.

## Competing interests

The authors declare that they have no conflicts of interest.

## Acknowledgements

We are grateful to the NASA scientists and staff for providing ATom data (https://espo.nasa.gov/atom/content/ATom) and
OMI data (https://daac.gsfc.nasa.gov/). The simulations were completed using a supercomputer (NEC SX-Ace and SX-Aurora TSUBASA) at NIES Japan. The surface observational data for model validation were obtained from the monitoring networks EANET (https://www.eanet.asia/) and EMEP (https://www.emep.int/).

**Financial support**

This research was supported by the Global Environment Research Fund (S–12 and S–20) of the Ministry of the Environment (MOE), Japan, and JSPS KAKENHI Grant Numbers: JP20H04320, JP19H05669, and JP19H04235. The study was also funded by the German Research Foundation (Deutsche Forschungsgemeinschaft; DFG) HALO-SPP 1294. The contributions from BS and KP were supported via the DFG grants PF 384/16, PF 384/17 und PF 384/19.

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
