# Peer review of "Implementation of HONO into the chemistry-climate model CHASER (V4.0): roles in tropospheric chemistry"

_Geoscientific Model Development, 2021_

## Author Comment (AC1)

Response to the Anonymous Referee #1 and #2's comments,

We thank the Anonymous Referees for our manuscript's thorough and constructive comments.

**Referee#1's comment:** Heterogeneous chemistry of HONO formation and sink in the atmosphere is one of the least quantified issues in tropospheric chemistry, which induce serious uncertainty in the global and regional CTM to predict O3 formation, CH4 lifetime, and so on. The present article implements the heterogeneous HONO chemistry into the chemistry-climate model CHASER to show the inclusion of HONO chemistry reduces the model bias against the measurements for PM2.5, NO3-/HNO3, NO2, OH, O3 and CO in the lower troposphere.

Since the importance of heterogeneous production and loss of HONO has rarely been treated by a global CTM, it is interesting and worthwhile to evaluate the role of the chemistry of HONO in a global scale, and the present study is a challenging effort toward the direction. The most serious problem of the present article, however, is that the effects of HONO chemistry in the global atmosphere are discussed in 3.1 and 3.2 without enough validation of the processes and assumed uptake coefficients for heterogeneous formation and loss of HONO. At the present stage of understanding of heterogeneous HONO chemistry, selection of appropriate processes and uptake coefficients to reproduce the HONO concentration in the urban and remote field measurements should be the starting point for the model discussion. I am afraid that the present article does not fulfill such requirement.

Therefore, I rather reject the present version of the paper for publication, but do encourage the authors to revise the paper considering the following comments and resubmit the paper after conducting appropriate recalculation.

**Author comment:** We are thankful for your dedicated time and interest in our manuscript. We acknowledged the shortcoming of this manuscript version. From the perspective of a global modeller, we tried to add the HONO chemistry into our global model CHASER to better predict the chemistry of O3, HOx, and CH4 lifetime, etc. This decade, HONO chemistry has received considerable interest, yet modelling studies for HONO on a global scale are still scarce. Implementation of HONO chemistry in a global model is a real challenge due to its significant uncertainties; the most relevant to this study is which one among various possible HONO formation mechanisms is dominant globally. The current manuscript version tried to give a fresh outcome on how the troposphere will respond to the basic mechanisms of HONO. Taking the chance to have the measured HONO levels within the free troposphere, we tried to reproduce the daytime HONO that was surprisingly high already for such an Asian coastal region.

We tried out your suggestion on a photochemical heterogeneous formation in an updated version this time. This new production resulted in better agreement with EMeRGe's HONO measurement. Therefore, the main new results are added in Sect. 3.1.2 and related places in the revised manuscript version, highlighted in yellow. We hope this version finds you more satisfactory since we also discussed the uncertainties of each potential HONO formation mechanism and its global effects (Sect. 3.2.1 and 3.2.3).

**Referee#1 comment:**

Specific suggestions for improving the paper:

Many field observations and regional CTMs have revealed that the concentrations of daytime HONO in urban area is much higher than expected by models considering only dark heterogeneous formation process (e.g., Lee et al., 2016; Lu yet al., 2018; quoted in the present paper). Since photochemical heterogeneous formation is now widely accepted to be important as a source of HONO, it should be taken into account as one of the important processes in the global model as well. The present paper discusses such photochemical process only later in 3.1.5.

Further, it has been generally accepted that every surface including soil dust and PM5, and the Earth's ground would be effective for the formation of HONO. These processes should be taken into account in the global model as well.

**Author comment:**

We agree that photochemical heterogeneous formation is an important source of HONO that should be coupled into modelling works of HONO. We also acknowledged that surface-related sources from Earth's ground, soils, and atmospheric particles might also be significant sources of HONO for dense and polluted regions. In this manuscript version, we added our new results of implementing such a photochemical heterogeneous HONO formation, which is the surface-catalyzed photolysis of HNO3 (HNO3 + hν → HONO) (R7), following the work by Lee et al. (2016). Our model accessed this photolysis at a rate 100 times faster than the gas-phase HNO3 photolysis (HNO3 + hν → OH + NO2). To access the assumption of this photolysis for aerosol nitrate and other aqueous surfaces and adsorbed HNO3 on ground surfaces, we conducted three sole simulations for (R7). The first run (namely JANO3-A) is for ground surfaces only, applying (R7) for the first vertical layer. The second and third runs (JANO3-B and JANO3-C) are for surfaces of grounds and all-kind aerosols, addressing (R7) for the model grid cells of particular surface area density ranges. Fortunately, the JANO3-B and JANO3-C cases successfully produce HONO at daytime for the height 0 ~ 2000 m during EMeRGe flights, although their effects in increasing NO2 and O3 worsened the model bias for these species. We tried to amend this problem by the combined cases of JANO3-B and JANO3-C with another promising case (maxST), in which the uptake coefficient of aerosol-uptakes of NO2 was raised to 0.1, resulting in more HONO during EMeRGe, but the reductions for NO2 and O3 were enhanced. The combined cases succeed in closing the gaps for HONO, NO2, and O3 levels with EMeRGe's measurement. The additional discussions of these extra simulations regarding the model's NOx-recycle issue, global HONO distribution and burden, and the effects to global oxidizing capacity were also added in the new version, highlighted in yellow.

**Referee#1 comment:**

As for the quantitative choice of uptake coefficients of heterogeneous processes, it is suggestive to parameterize them to reproduce the observational HONO concentration in the flied. The sensitivity of uptake coefficient of the heterogeneous formation processes would be high in the boundary layer in polluted areas, validation in urban areas should be performed at first by the global model comparing with the results of regional models. Then, validation for vertical profiles by aircraft measurements of EMeRGe and Atom1 should be made before the discussion of the effect of HONO chemistry in the lower troposphere in general.

**Author comment:**

We made rearrangements for the manuscript order as per your suggestion. Please find the new construction in our updated version in section 3.2 in the evaluation and verification order as for EMeRGe → Atom → Mirai → EANET/EMEP. In addition, the comparison for TCO with OMI was moved to before the discussion on global impacts, as TCO changes could be seen as a global effect on ozone.

**Referee#1 comment:**

Although the present paper refers limited laboratory studies on the heterogeneous dark reactions of HONO formation and loss on specific aerosol such as soot, the uncertainties are more than an order of magnitude, and it is not appropriate to select particular values for the "standard" run as in Table 2. Considering the large uncertainty of the uptake coefficients at this stage, the empirical approach mentioned above is suggested to be followed in the future paper.
over

**Author comment:**

We are sorry for the large uncertainties in the particular values of uptake coefficients used in the "standard" run. It is the first try to perform HONO chemistry in our model. We simplified the related configurations at the least computational cost, using the averaged uptake coefficients and the most basic pathways of gaseous, heterogeneous reactions of HONO and direct emission. This manuscript tells the first results of this approach to HONO chemistry, although we tried various sensitivity simulations for amplifiyng the existing mechanisms. Adding the new mechanisms such as photochemical heterogeneous processes is complex; thus, we did not conduct any before this response. In the revised manuscript version, we sincerely thank you for checking again our new results, which resulted from adding the heterogeneous photolysis of HNO3, one of your valuable suggestions.

We hope this version can share the current knowledge of global HONO chemistry.

**Response to Anonymous Referee #2's comment,**

**Referee#2's comment:** A review report for manuscript entitled "Implementation of HONO into the chemistry-climate model CHASER (V4.0): roles in tropospheric chemistry" by Phuc et al., 2021.

The study investigates the impacts of including HONO three formation paths (gas phase, heterogeneous, and emissions sources) on the levels of PM2.5, SO42-, HNO3, CO, O3, in the model compared to measurements from different platforms. However, the study does not account for major HONO sources via night and daytime heterogeneous NO2 conversion on-ground sources, and nitrate photolysis. In addition, the study presents the results of comparison but without a sufficient or reasonable explanation of these differences. This is a major issue, and the authors should carefully review the article and provide adequate clarifications accounting for HONO impact on each species beyond increase or decrease in species concentrations.

**Author's comment:** We sincerely thank the referee for your time and devotion to pointing out our manuscript's shortcomings. To improve the manuscript, we will specifically clarify each point.

**Referee#2's comment:** Specifics

Abstract: remove "for the first time" since HONO has been parameterized in several previous studies (e.g., Elshorbany et al., 2012; Zhang et al., 2021).

**Author's comment:** We removed the ambiguous phrase in the revised manuscript as suggested (**line 12**).

**Referee#2's comment:** Page 1, line 24: the 51% contribution of heterogeneous NO2 to HONO formation on aerosol, and emissions sources are very high compared to current literature (e.g., Zhang et al., 2021).

**Author's comment:** Zhang et al. (2021) implemented in the CMAQv5.3 3D-model six homogeneous reactions and five distinct heterogeneous reactions for day and nighttime. They included HONO emissions from vehicles and not from soil. Their simulations resulted in dominant contribution of heterogeneous production from NO2 adsorbed on the ground (~75 %) of the total HONO production for the haze days in Beijing (China). However, they also found that the second dominant contributor to HONO production is heterogeneous production from NO2 adsorbed on aerosol surfaces. Our study did not include NO2 conversion on ground surfaces, while HONO emission from vehicles, combustion, and soils are all considered. That is why the contributing portions of NO2's aerosol-uptakes and HONO emission are higher in our study than in Zhang's calculation.

**Referee#2's comment:** Page 1, line 30: Why does reducing NOx (NO2+NO) reduce the atmospheric oxidation capacity? For instance, reducing NO2 would increase OH in high NOx conditions. Please clarify.

**Author's comment:** for high $NO_x$ conditions (e.g. China), reducing NO2 increases regional OH and $O_3$ levels (Fig. 12, first and third rows). However, for the larger part of the globe (non-polluted regions, higher altitudes), the NOx-deficit environments caused by $NO_2$ uptakes on clouds/aerosols (mainly on clouds) restricts O3 formation and OH radical formation (via O3 photolysis or via $HO_2+NO \rightarrow NO_2+OH$ reaction) at these environments (explanation is given in **lines 729-732** in the new version). Therefore, we added an explanation at **line 32** as well.

**Referee#2's comment:** Page 6, line 169: I don't see the value of the OLD simulations which assumed no HONO chemistry since all models have at least gas-phase HONO chemistry (OH+NO=HONO;HONO=OH+NO; and HONO+OH).

**Author's comment:** we added a description for the OLD case, highlighted in **lines 137-140** and **202-204**. In the OLD case, the model excludes HONO species and HONO chemistry, and there are no gas-phase HONO reactions in the OLD case (Table 1). Results for the OLD are also exhibited in sub-sections in 3.1 (Figures 2 to 8).

**Referee#2's comment:** Page 6, table 2: How is the heterogeneous loss of HONO (R6) leads to NO?

**Author's comment:** The loss process of HONO occurs on surfaces of ice, liquid and aqueous sulfate aerosols via the reaction $HONO + H_2O \rightarrow NO^- + H_3O^+$, since HONO uptake by liquid water probably involves hydrolysis (Revised manuscript, line 73). Our model simplified this hydrolysis as HONO $\rightarrow$ NO (R6). NO will be converted to NO2 in a few hours in the atmosphere. We have checked this reaction with (R6) being HONO $\rightarrow$ NO2 and there is no big difference in the results.

**Referee#2's comment:** Page 7, line 175: I suggest you differentiate NO from NO2 in model calculations throughout the paper.

**Author's comment: Line 208.** This line is about the equation Eq.(1) to determine the effects of each mechanism on atmospheric species (OH, O3, NOx, CO). We agree that, for HONO chemistry, NO2 is an important precursor of HONO that should be separately investigated from NO. Compared with ATom and EMeRGe data, NO2 measurement is available, so simulated NO2 data is used in these parts. However, only NOx data is available for other comparisons (EANET, EMEP). Therefore, we use simulated NOx data to compare measurements at EANET and EMEP stations. Also, NOx is used to calculate global effects, as NOx can better represent an important tropospheric species than NO2 solely.

**Referee#2's comment:** Page 9, line 230: Please report the calculated surface aerosol density for each aerosol type.

**Author's comment: Line 267-276.** We added the calculated SAD values for each aerosol type per your suggestion.

**Referee#2's comment:** Page 10, lines 238-247: Please elaborate on "O3-reducing effects of HONO chemistry".
The authors should explain how HONO would reduce O3 when its photolysis is a source of OH, supposedly increasing the oxidation capacity?

**Author's comment:** Although HONO photolysis (R1) is a source of OH, the calculation in STD case shows the OH and $O_3$ increases only occur at the surface of polluted sites. For remote regions, the $NO_2$ conversion to HONO and $HNO_3$ (R4) becomes a removal pathway for $NO_x$, thus restricting the formation of $O_3$ and OH for the larger part of the troposphere via lacking oxygen atom from $NO_2$ photolysis. We added elaboration on "$O_3$-reducing effects of HONO chemistry" from **line 669~** (revised manuscript)

**Referee#2's comment:** Page 11, line 262-264: The authors are advised to explain the causal factors leading to increase or decrease in the impacted species (PM2.5, SO42-, HNO3, O3..etc) rather than stating the numbers.

**Author's comment: Line 539-546**. We added the explanation for increase or decrease in HNO3, NO3, PM2.5, SO42-, and referred to the statistical values in Table 7 along with the explanation.

**Referee#2's comment:** Page 11, line 285: Again, an explanation of causal factors is missing.

**Author's comment: Line 559.** We added the explanation highlighted in yellow.

**Referee#2's comment:** Page 14, line 344: Again, the authors should explain how the inclusion of gas-phase HONO sources led to increased CO in some regions but decreased CO in other regions…

**Author's comment: Lines 489-495.** We added the explanation and Figure S14 (supplement) to support the understanding of O3-CO chemistry during *Mirai* observation.

**Referee#2's comment:** Page 17, line 382 and Figure 7: Figure 7 does not show the vertical profiles of simulated HONO. I also don't think that HONO will have any impacts at 200 hPa??

**Author's comment: Line 418**. We don't have the available HONO measurement from ATom, so Figure 5 (Figure 7 in the last version) does not show the simulated vertical HONO level, instead of showing only model biases and changes in $NO_2$, $O_3$, OH, CO, induced by HONO chemistry. We changed the text to clarify this. We agree that HONO chemistry's impacts mainly stay in the lower and middle troposphere (Figure 12 (i-p)).

**Referee#2's comment:** Page 21, lines 445: HONO values at these heights are extremely high. HONO values of 70 ppt at 2000m are almost impossible given its ground-based sources and its fast photolysis. Authors should show some evidence that these numbers are reasonable.

**Author's comment: Lines 328-332**. These values (mean 70 ppt HONO at 2000m) is from the EMeRGe measurement (Andrés Hernández et al., 2021). Our simulation in the STD case also can not reproduce this high level of HONO. Yes, we agree that the measured HONO level measured in EMeRGe for height 2,000 m ± 500 m (10-115 ppt) is surprisingly high. Wang et al. (2019) reported vertical HONO profile measured by a MAX-DOAS instrument at a station in the North China Plain and presented only < 100 ppt of measured HONO in May 2016 and averaged of < 30 ppt HONO during April to June 2016 at this height. We added a statement and reference for this issue in the revised manuscript.

References used by author's comments:

Andrés Hernández, M. D., Hilboll, A., Ziereis, H., Förster, E., Krüger, O. O., Kaiser, K., Schneider, J., Barnaba, F., Vrekoussis, M., Schmidt, J., Huntrieser, H., Blechschmidt, A.-M., George, M., Nenakhov, V., Klausner, T., Holanda, B. A., Wolf, J., Eirenschmalz, L., Krebsbach, M., Pöhlker, M. L., Hedegaard, A. B., Mei, L., Pfeilsticker, K., Liu, Y., Koppmann, R., Schlager, H., Bohn, B., Schumann, U., Richter, A., Schreiner, B., Sauer, D., Baumann, R., Mertens, M., Jöckel, P., Kilian, M., Stratmann, G., Pöhlker, C., Campanelli, M., Pandolfi, M., Sicard, M., Gomez-Amo, J. L., Pujadas, M., Bigge, K., Kluge, F., Schwarz, A., Daskalakis, N., Walter, D., Zahn, A., Pöschl, U., Bönisch, H., Borrmann, S., Platt, U., and Burrows, J. P.: Overview: On the transport and transformation of pollutants in the outflow of major population centres – observational data from the EMeRGe European intensive operational period in summer 2017, Atmos. Chem. Phys. Discuss. [preprint], doi: 10.5194/acp-2021-500, in review, 2021.

Lee, J. D., Whalley, L. K., Heard, D. E., Stone, D., Dunmore, R. E., Hamilton, J. F., Young, D. E., Allan, J. D., Laufs, S., Kleffmann, J.: Detailed budget analysis of HONO in central London reveals a missing daytime source. Atmos. Chem. Phys., 16, 2747–2764. doi: 10.5194/acp-16-2747-2016, 2016.

Sincerely,

On behalf of all co-authors,
Phuc T. M. Ha.

---

## Author Comment (AC2)

**Implementation of HONO into the chemistry-climate model CHASER** (V4.0): roles in tropospheric chemistry**

Phuc T. M. Ha1, Yugo Kanaya2, Fumikazu Taketani2, Maria Dolores Andrés Hernández3, Benjamin Schreiner4, Klaus Pfeilsticker4, Kengo Sudo1,2

[revised manuscript text omitted]

| Base model                              | MIROC4.5 AGCM                                                                                                                                                                                                                                                                                                                                                                                                                                                                                                                                                             |
|-----------------------------------------|---------------------------------------------------------------------------------------------------------------------------------------------------------------------------------------------------------------------------------------------------------------------------------------------------------------------------------------------------------------------------------------------------------------------------------------------------------------------------------------------------------------------------------------------------------------------------|
| Spatial resolution                      | Horizontal, T42 (2.8° × 2.8°); vertical, 36 layers (surfaces approx. 50 km)                                                                                                                                                                                                                                                                                                                                                                                                                                                                                               |
| Meteorology
(u, v, T)                | Nudged to the NCEP2 FNL reanalysis                                                                                                                                                                                                                                                                                                                                                                                                                                                                                                                                        |
| Emission
(anthropogenic,
natural) | Industry traffic, Vegetation Ocean
Biomass burning specified by MACC reanalysis                                                                                                                                                                                                                                                                                                                                                                                                                                                                                        |
| Aerosol                                 | BC/OC, sea-salt, and dust
BC ageing with SO x /SOA production                                                                                                                                                                                                                                                                                                                                                                                                                                                                                               |
| Chemical process                        |  <li>94 chemical species, 263 chemical reactions (gas phase, liquid phase, non-uniform Ox-NOx-HOx-CH4-CO chemistry with VOCs SO2, DMS oxidation (sulfate aerosol simulation) SO4-NO3-NH4 system and nitrate formation Formation of SOA BC ageing</li> <li>(+) Heterogeneous reactions: 8 reactions of N2O5, HO2, RO2; constant uptake coefficients (γ) on types of aerosols (Ice, Liquid, Sulfate, Sea salt, Dust, OC)</li>  |

[revised manuscript text omitted]

---

## Author Comment (AC3)

**Supplements**

| Country           | Station (ID number)                                                                  |
|-------------------|--------------------------------------------------------------------------------------|
| Cambodia          | PhnomPenh (31)                                                                       |
| China             | Jinyunshan (48), Hongwen (14)                                                        |
| Indonesia         | Jakarta (18), Serpong (36), Bandung (1)                                              |
| Japan             | Rishiri (33), Ochiishi (26), Tappi (38), Sado-seki (34), Happo (10), Ijira (15), Oki |
|                   | (28), Banryu (3), Yusuhara (45), Hedo (11), Ogasawara (27), Tokyo (40)               |
| Lao               | Vientiane (42)                                                                       |
| Malaysia          | Petaling Jaya (30), Tanah Rata (37), Danum Valley (7)                                |
| Mongolia          | Ulaanbaatar (41), Terelj (39)                                                        |
| Myanmar           | Yangon (43), Mandalay (50)                                                           |
| Philippines       | Metro Manila (22), Mt. Sto. Tomas (24)                                               |
| Republic of Korea | Kanghwa (19), Cheju (Kosan) (5), Imsil (16)                                          |
| Russia            | Mondy (23), Listvyanka (21), Irkutsk (17), Primorskaya (32)                          |
| Thailand          | Bangkok (2), Samutprakarn (47), Pathumthani (29), Khanchanaburi (20), Chiang         |
|                   | Mai (6), Sakaerat (35), Nai Mueang (25), Chang Phueak (46), Si Phum (49)             |
| Vietnam           | Hanoi (8), Hanoi (Relocated) (9), Hoa Binh (13), Can Tho (4), Ho Chi Minh (12),      |
|                   | Yen Bai (44)                                                                         |

Table S 1: Lists of EANET stations grouped by their countries with ID number as in Error! Reference source not found..

5 Table S 2: Model comparison with ATom1 flights, calculated for all flight, and for North Pacific (NP) region: no outlier detection is applied. N is number of available data for each calculation, R is the correlation coefficients. *R* and bias of the STD run are shown as bold if better than that of the OLD run. Unit of bias is ppt for NO2, OH, ppb for O3, CO.

|            | NO 2 | NO 2 | O 3 | 03     | ОН     | OH     | CO     | СО      |
|------------|-----------------|-----------------|------------|--------|--------|--------|--------|---------|
|            |                 | (NP)            |            | (NP)   |        | (NP)   |        | (NP)    |
| N          | 29,509          | 2,283           | 29,204     | 2,246  | 7,601  | 608    | 27,467 | 2,172   |
| R (STD)    | 0.730           | 0.621           | 0.751      | 0.609  | 0.579  | 0.407  | 0.659  | 0.596   |
| R (OLD)    | 0.697           | 0.306           | 0.752      | 0.598  | 0.584  | 0.374  | 0.643  | 0.596   |
| bias (STD) | -11.277         | 0.588           | 11.637     | 8.471  | -0.038 | -0.003 | 1.698  | -1.713  |
| bias (OLD) | -6.940          | 4.450           | 15.025     | 13.050 | -0.015 | 0.015  | -7.521 | -12.393 |

**10 **Table S 3: Additional sensitivity runs for the EMeRGe comparison.**

The AIRC case aims to evaluate source of HONO from aircraft emissions of HONO using a HONO/NOx emission factor of 0.4. In the EMx8 case, the HONO/NOx emission factor is amplified up to 0.8 (=0.1 in STD case) to emphasize the sensitivity of HONO's direct source from the ground layer, especially from soils (Oswald et al., 2013). In the GRx8 case, the rate constant of (R2) is eightfold to increase homogeneous HONO production, given that daytime missing HONO could relate to other gas-phase formations (Romer et al., 2018; Li et al., 2014). The factor 8 applied in EMx8 and GRx8 cases are selected after testing with factors 2 and 4, aiming for simulations to agree with the measurements.

| No. | Simulation ID | Description                                                     | Note               |
|-----|---------------|-----------------------------------------------------------------|--------------------|
| 1   | AIRC          | aircraft HONO emission = 0.4% aircraft NO x emission | Not applied in STD |
| 2   | GRx8          | Rate (R2) × 8                                                   |                    |
| 3   | EMx8          | $EM(HONO) = 0.8 NO_x$ emission                                  | = 0.1 in STD       |

15

**Table S 4: Tables of correlation coefficient (R) and model biases against EMeRGe measurements for HONO.**

"Alt." columns show altitude ranges ( $\pm 500$  m). "N" column show the numbers of hourly-averaged values calculated for each altitude range. Left table: darker colours represent higher absolute values of *R* (closer to  $\pm 1$ ). Right table: lighter colours show smaller model biases (closer to 0). The darkness of blues (negative values) and reds (possitive values) are scaled to  $\pm 1$  for *R* and  $\pm$ maximum values of each row for biases. Unit of biases is ppt for HONO and NO2, ppb for O3 and CO.

|      |      | R(HONO) |       |       |       |       |       |               |         |         |                   |                   |        | Bi    | as(HON | 0)     |       |        |               | 20      |         |                   |                   |
|------|------|---------|-------|-------|-------|-------|-------|---------------|---------|---------|-------------------|-------------------|--------|-------|--------|--------|-------|--------|---------------|---------|---------|-------------------|-------------------|
| Alt. | N    | STD     | GRx8  | EMx8  | AIRC  | maxST | ratR4 | ratR4
+CLD | JANO3-B | JANO3-C | maxST+
JANO3-B | maxST+
JANO3-C | NEW    | GRx8  | EMx8   | AIRC   | maxST | ratR4  | ratR4
+CLD | JANO3-B | JANO3-C | maxST+
JANO3-B | maxST+
JANO3-C |
| 0    | 970  | -0.23   | -0.39 | -0.29 | -0.27 | -0.17 | -0.22 | -0.21         | 0.63    | 0.62    | 0.64              | 0.63              | -112.5 | -94.1 | -102.7 | -112.2 | -70.3 | -106.1 | -102.9        | -21.7   | -17.6   | 155.0             | 154.9             |
| 1000 | 1714 | 0.49    | 0.36  | 0.51  | 0.44  | 0.56  | 0.24  | 0.24          | 0.36    | 0.37    | 0.48              | 0.48              | -105.3 | -95.5 | -94.2  | -105.6 | -71.7 | -99.8  | -96.1         | -47.8   | -40.8   | 65.9              | 72.3              |
| 2000 | 1538 | 0.31    | 0.47  | 0.38  | 0.36  | 0.47  | 0.12  | 0.07          | 0.47    | 0.40    | 0.41              | 0.39              | -64.1  | -62.9 | -64.1  | -64.4  | -61.8 | -63.3  | -62.8         | -53.1   | -45.6   | -32.5             | -23.6             |
| 3000 | 2296 | 0.16    | 0.05  | 0.11  | 0.11  | -0.03 | 0.13  | 0.05          | 0.34    | 0.28    | 0.18              | 0.26              | -44.2  | -42.8 | -44.1  | -44.2  | -43.2 | -43.9  | -43.7         | -38.7   | -30.2   | -27.7             | -16.2             |
| 4000 | 192  | -0.17   | -0.24 | -0.08 | -0.11 | 0.28  | -0.11 | -0.04         | 0.08    | -0.14   | 0.36              | 0.30              | -26.0  | -24.3 | -25.8  | -26.0  | -25.6 | -25.7  | -25.4         | -23.8   | -17.9   | -21.2             | -14.4             |
| 5000 | 836  | 0.04    | 0.03  | 0.14  | 0.21  | 0.53  | 0.19  | 0.75          | 0.17    | -0.22   | 0.49              | 0.06              | -18.9  | -17.3 | -18.8  | -19.0  | -18.5 | -18.8  | -18.3         | -17.7   | -12.8   | -15.6             | -9.6              |
| 6000 | 506  | -0.01   | 0.02  | -0.03 | 0.03  | 0.11  | -0.03 | 0.05          | 0.10    | -0.26   | 0.16              | -0.12             | -5.0   | -2.9  | -4.6   | -5.1   | -4.9  | -4.8   | -4.8          | -4.1    | 2.5     | -3.9              | 2.2               |
| 7000 | 76   | -0.31   | -0.33 | -0.31 | -0.33 | -0.30 | -0.30 | -0.30         | -0.29   | -0.29   | -0.27             | -0.22             | -4.1   | 0.7   | -3.5   | -4.3   | -4.1  | -3.8   | -3.7          | -2.3    | 1.5     | -2.0              | 1.7               |
| 8000 | 44   | -0.67   | -0.64 | -0.64 | -0.64 | -0.64 | -0.68 | -0.67         | -0.62   | -0.65   | -0.63             | -0.59             | -2.8   | 2.9   | -1.9   | -2.7   | -2.7  | -2.5   | -2.5          | -1.5    | 2.9     | -1.2              | 3.5               |

| Simulation ID    | CH 4 lifetime | lifetime Abundances of tropospheric |                     |        |          |  |  |  |
|------------------|--------------------------|-------------------------------------|---------------------|--------|----------|--|--|--|
|                  | (yr)                     | NO x                     | O 3      | СО     | HONO     |  |  |  |
|                  |                          | (TgN)                               | (TgO 3 ) | (TgCO) | (TgN)    |  |  |  |
| OLD              | 9.09                     | 0.119                               | 408.79              | 327.20 |          |  |  |  |
| STD              | 10.28                    | 0.094                               | 388.21              | 354.57 | 1.40     |  |  |  |
| maxST            | 14.54                    | 0.048                               | 323.80              | 425.31 | 7.79     |  |  |  |
| ratR4+CLD        | 9.60                     | 0.108                               | 390.34              | 337.68 | 3.18     |  |  |  |
| JANO3-A          | 10.05                    | 0.096                               | 391.11              | 349.91 | 1.45     |  |  |  |
| JANO3-B          | 7.60                     | 0.116                               | 426.89              | 292.29 | 2.02     |  |  |  |
| JANO3-C          | 5.39                     | 0.153                               | 477.48              | 237.59 | 2.93     |  |  |  |
| maxST+JANO3-B    | 10.20                    | 0.057                               | 351.27              | 357.27 | 12.64    |  |  |  |
| maxST+JANO3-C    | 6.44                     | 0.084                               | 408.69              | 268.74 | 17.13    |  |  |  |
| Effects          |                          |                                     |                     |        |          |  |  |  |
|                  |                          | vs OLD                              |                     |        |          |  |  |  |
| By STD           | +13.05                   | -20.40                              | -5.03               | +8.36  |          |  |  |  |
| by maxST         | +50.65                   | -55.44                              | -17.84              | +37.02 | +634.51  |  |  |  |
| by ratR4+CLD     | +5.60                    | -8.57                               | -4.51               | +3.20  | +129.94  |  |  |  |
| By JANO3-A       | +10.57                   | -18.97                              | -4.32               | +6.94  | +3.42    |  |  |  |
| By JANO3-B       | -16.39                   | -2.49                               | +4.43               | -11.06 | +44.2    |  |  |  |
| By JANO3-C       | -40.74                   | +28.89                              | +16.08              | -32.41 | +108.7   |  |  |  |
| By maxST+JANO3-B | +12.21                   | -52.10                              | -14.07              | +9.19  | +802.86  |  |  |  |
| By maxST+JANO3-C | -29.15                   | -29.41                              | -0.02               | -17.87 | +1123.57 |  |  |  |

Table S 5: CH4 lifetime and tropospheric abundances for NOx, O3, CO, and HONO and their changes by HONO chemistry in sensitivity cases.

---

## Referee Report (RR1)

Reviewer #1 comments to the revised manuscript

The revised manuscript has been much improved by the intensified validation of the model by the observed values obtained in EMeRGe. The present version of the article is basically acceptable for publication and the following comments are for improving the paper to an advanced stage.

The article will be logically more understandable if the validation of the model and the discussion are presented in the following order.

1. Comparison of the measurement data and the CHASER simulation for HONO/$NO_x$ in the urban/suburban boundary layer, where aerosol concentration is high and cloud contribution may be negligible.

   There are a few numbers of papers reporting high HONO in urban/suburban area in China and US, e.g.,

   > Lee et al., Atmos. Chem. Phys., 16, 2747-2764, 2016.
   > Ye et al., Atmos. Chem. Phys., 18, 9107-9120, 2016.
   > Zheng et al., Atmos. Chem. Phys., 20, 5457-5475, 2020.
   > Xue, et al., Atmos. Chem. Phys., 22, 1035-1057, 3149-3167, 2022.

   Please show the comparison of the measured and model simulated values for HONO/$NO_x$ using the best selected common $\gamma$-values for heterogeneous HONO formation processes on aerosol- and ground-surfaces including the heterogeneous photochemical HONO formation.

2. Next, show the comparison of measurement data of EMeRGe and model simulation demonstrating that the inclusion of heterogeneous formation of HONO on cloud water improves the agreement. Discuss the relative importance of cloud surface process for HONO formation in the free troposphere in a global scale.

3. Decrease of $HO_x$ and $O_3$ formation by the inclusion of HONO formation processes in the free troposphere have been discussed, which is against the general understanding that HONO formation increases $HO_x$ concentration and $O_3$ production in urban area. If it is ascribed to the situation under low concentration region of $NO_x$, threshold concentration of $NO_x$, where positive to negative contribution of HONO formation to the oxidant formation will occur, should be discussed.

over

---

## Author Response (AR2)

Response to the Anonymous Referee #1 and #2's Reports,

We thank the Anonymous Referees for our manuscript's thorough and constructive comments.

**Referee's suggestion:**

The revised manuscript has been much improved by the intensified validation of the model by the observed values obtained in EMeRGe. The present version of the article is basically acceptable for publication and the following comments are for improving the paper to an advanced stage.

1. Comparison of the measurement data and the CHASER simulation for HONO/NOx in the urban/suburban boundary layer, where aerosol concentration is high and cloud contribution may be negligible. There are a few numbers of papers reporting high HONO in urban/suburban area in China and US, e.g.,

Lee et al., Atmos. Chem. Phys., 16, 2747-2764, 2016.

Ye et al., Atmos. Chem. Phys., 18, 9107-9120, 2016.

Zheng et al., Atmos. Chem. Phys., 20, 5457-5475, 2020.

Xue, et al., Atmos. Chem. Phys., 22, 1035-1057, 3149-3167, 2022.

Please show the comparison of the measured and model simulated values for HONO/NOx using the best selected common γ-values for heterogeneous HONO formation processes on aerosol- and ground-surfaces including the heterogeneous photochemical HONO formation.

**Author's response:**

We thank the referee for their encouragement and the suggestion of a more comprehensive order for our manuscript.

However, we may preserve the order of the verification part (Sect. 3.1), beginning with the model comparison with EMeRGe in the free troposphere to utilize the HONO-measured data on HALO. This very inclusive dataset drives our effort to make various trials on different HONO production mechanisms for understanding HONO chemistry. Only through this comparison did we notice that the standard and sensitivity simulations (Table 3) were insufficient to explain the measurement along EmeRGe. We also noticed that additional cases (Table 5), with our trials for various mechanisms, needed to explain the measurement. Several cases were successful for different conditions along EMeRGe flights, while other cases were more efficient for other comparisons (ATom, Mirai, EANET, EMEP). The order of our comparison hence started from EmeRGe (free troposphere in coastal East-Asian region) to ATom (free troposphere in the near and far-coastal regions), Mirai (coastal surface sea environment), then EANET, EMEP (continental ground-based observations). Besides EANET, EMEP, we added a comparison for HONO, NO2, $O_3$ during the summer of 2018 of Mt. Tai (China) by reproducing data in Xue et al.'s report (Xue et al., 2022) to compare with simulated concentrations in our model **(Lines 529 - 571)**. We then proposed the most potential HONO production mechanisms for the specific environments being compared.

Comparing CHASER simulation and the measurement in the urban/suburban boundary layer from EANET, EMEP, and Xue's data (Sect 3.1.5), we acknowledge that aerosol concentration is high for these environments. We discussed that the relative importance of aerosol uptakes in the sensitivity of HONO formation for the summit station was higher than those for the foot station **(Lines 541-544)**, and the sensitivity of $O_3$ formation **(Lines 559-562)**.

**Referee's suggestion:**

2. Next, show the comparison of measurement data of EMeRGe and model simulation demonstrating that the inclusion of heterogeneous formation of HONO on cloud water improves the agreement.

**Author's response:**

In the impact parts (Sect. 3.2), we added detailed discussions for aerosols and cloud effects for specific environments being compared in Sect. 3.1, concluding that adding cloud effects improved agreement in the free troposphere (generally improved for comparison with ATom, partially improved in CO simulation by cloud effect) **(lines 807-830)**.

**Referee's suggestion:**

Discuss the **relative importance of cloud surface process for HONO formation in the free troposphere in a global scale**.

**Author's response:**

The relative importance of aerosol effects on HONO formation, including particle NO3-photolysis (JANO3-B and maxST+JANO3-B cases), was discussed in Sect. 3.2 **(Lines 823-830)**.

The case JANO3-B and the combined case maxST+JANO3-B enhanced $NO_2$'s aerosol-uptakes (R4, R5) and $NO_3^-$ photolysis (R7) on the ground and aerosol surfaces (SAD threshold was set as $10^{-6}$ - $10^{-4}$ $cm^2$ $cm^{-3}$ to exclude this photolysis on cloud particles). However, in these simulations, the cloud effects for the uptakes of $NO_2$ and HONO (R4, R6) still be active. Thus, we used this simulation to discuss the sensitivity of HONO formation and relevant chemistry to aerosol effects (not the magnitude of aerosol-effect themselves).

Regarding the relative importance of the cloud surface process for HONO formation and global oxidizing power in the free troposphere, we added in Sect. 3.2 the discussion on the magnitude of cloud effects (= GR+HR(cld) – GR; see configurations for simulations in Table 3) in **Lines 807-822**. Also, the sensitivity of HONO formation and oxidizing chemistry to cloud surface was also added in Sect. 3.2, via enhancement of (R4) on cloud surface in the ratR4+CLD case ($\gamma_{liq.}$(R4) = 0.01) and via SAD threshold for cloud in JANO3-C cases.

**Referee's suggestion:**

3. Decrease of HOx and O3 formation by the inclusion of HONO formation processes in the free troposphere have been discussed, which is against the general understanding that HONO formation increases HOx concentration and O3 production in urban area. If it is ascribed to the situation under low concentration region of NOx, threshold concentration of NOx, where positive to negative contribution of HONO formation to the oxidant formation will occur, should be discussed.

**Author's response:**

To draw a picture of the correlation between NOx concentration and HONO's impacts on oxidizing chemistry (OH, O3, CO, CH4), we added discussion in Sect. 3.2. In **lines 834-843**, the tendency sensitivity of HONO's impact on oxidant species (OH and $O_3$) to $NO_x$ concentration was added. In **lines 856-872**, the discussion on the positive to a negative contribution of HONO formation mechanisms to the oxidant formation will occur was added.

**Review report from Referee #2:**

A second review report on "Implementation of HONO into the chemistry-climate model CHASER (V4.0): roles in tropospheric chemistry" by Phuc et al., 2021. My responses are based on the authors' responses and the revised version of the manuscript.

Since this is a revised manuscript, I don't think the manuscript is publishable since the authors still didn't account for the ground sources, which are the major HONO source, and because they calculate NO2 reduction of 20% which I don't believe a correct number since global HONO/NOx ratio can't exceed 4% even if all sources are accounted for.

**Author's response:**

Thank you so much for your time in reviewing our manuscript thoroughly once more. We understand that the revised manuscript did not align with your thinking.

In the revised manuscript, we tried to account for the ground source of $NO_3^-$ photolysis (JANO3-A case), which was suggested by Lee et al. (2016). The contribution of particle $NO_3^-$ photolysis on the ground surface (JANO3-A case) was addressed in comparison with EmeRGe (Sect. 3.1.2) and once more in the impact part (Sect. 3.2.3; **lines 847-849**).

The calculated $NO_2$ reduction of 20% by the STD case is naturally sceptical. After adding particle-phase $NO_3^-$ photolysis to the model, the global impacts on tropospheric $NO_x$ levels ranged in different tendencies (Figure 15). Only in the JANO3-C case could the strong NO3-photolysis on ground, aerosols, and cloud surfaces (SAD ( 10 μm2cm-3) increase the global $NO_x$ abundance. In most of the other simulations, the $NO_x$ abundance was still reduced due to its removal via $NO_2$ uptakes, followed by OH and $O_3$ reductions. Hence, we concluded that the impact tendencies were similar to the STD case as $NO_2$ uptakes are still an active removal process for $NO_x$, but the magnitude of global changes in tropospheric species are still with high uncertainties.

**Reference**

Xue, C., Ye, C., Kleffmann, J., Zhang, C., Catoire, V., Bao, F., Mellouki, A., Xue, L., Chen, J., Lu, K., Zhao, Y., Liu, H., Guo, Z., and Mu, Y.: Atmospheric measurements at Mt. Tai – Part I: HONO formation and its role in the oxidizing capacity of the upper boundary layer, Atmos. Chem. Phys., 22, 3149–3167, https://doi.org/10.5194/acp-22-3149-2022, 2022a.

Sincerely,

On behalf of all co-authors,

Phuc T. M. Ha.